# Cryo-EM structure of severe fever with thrombocytopenia syndrome virus

Shouwen Du [1,2,3,11], Ruchao Peng [4,11], Wang Xu[5,11], Xiaoyun Qu[1,6,11], Yuhang Wang[2], Jiamin Wang [5], Letian Li[5], Mingyao Tian [5], Yudong Guan[2], Jigang Wang[2], Guoqing Wang [3], Hao Li [7], Lingcong Deng[5], Xiaoshuang Shi[5], Yidan Ma[5], Fengting Liu[2], Minhua Sun[1], Zhengkai Wei[1], Ningyi Jin[5], Wei Liu[7] ✉, Jianxun Qi [8,9] ✉, Quan Liu [1,3,10] ✉, Ming Liao[1,6] ✉ & Chang Li [3,5] ✉

The severe fever with thrombocytopenia syndrome virus (SFTSV) is a tick-borne human-infecting bunyavirus, which utilizes two envelope glycoproteins, Gn and Gc, to enter host cells. However, the structure and organization of these glycoproteins on virion surface are not yet known. Here we describe the structure of SFTSV determined by single particle reconstruction, which allows mechanistic insights into bunyavirus assembly at near-atomic resolution. The SFTSV Gn and Gc proteins exist as heterodimers and further assemble into pentameric and hexameric peplomers, shielding the Gc fusion loops by both intra- and inter-heterodimer interactions. Individual peplomers are associated mainly through the ectodomains, in which the highly conserved glycans on N914 of Gc play a crucial role. This elaborate assembly stabilizes Gc in the metastable prefusion conformation and creates some cryptic epitopes that are only accessible in the intermediate states during virus entry. These findings provide an important basis for developing vaccines and therapeutic drugs.

Since SFTSV was identified in central China in 2009, epidemics of SFTSV have occurred in many East Asian countries[1–4], and escalating incidences of infection have been reported in recent years[5], suggesting the potential to cause large scale pandemics in human populations. The life cycle and underlying mechanism of SFTSV transmission in humans have not been well understood. It is thought to transmit mainly through ticks as the vector, though human-to-human transmissions were also reported through direct contact with body fluids of infected patients[6,7]. Unfortunately, no vaccines or specific drugs are available so far, and SFTSV has been listed as one of the priority target

[1]Key Laboratory of Livestock Disease Prevention of Guangdong Province, Scientific Observation and Experiment Station of Veterinary Drugs and Diagnostic Techniques of Guangdong Province, Ministry of Agriculture and Rural Affairs, Institute of Animal Health, Guangdong Academy of Agricultural Sciences, Guangzhou, China. [2]The First Affiliated Hospital (Shenzhen People's Hospital), Southern University of Science and Technology, Shenzhen, China. [3]Department of Infectious Diseases and Center for Infectious Diseases and Pathogen Biology, Key Laboratory of Organ Regeneration and Transplantation of the Ministry of Education, State Key Laboratory for Diagnosis and Treatment of Severe Zoonotic Infectious Diseases, Key Laboratory for Zoonosis Research of the Ministry of Education, The First Hospital of Jilin University, Changchun, China. [4]Department of Biochemistry and Biophysics, Perelman School of Medicine, University of Pennsylvania, Philadelphia, PA, USA. [5]Research Unit of Key Technologies for Prevention and Control of Virus Zoonoses, Chinese Academy of Medical Sciences, Changchun Veterinary Research Institute, Chinese Academy of Agricultural Sciences, Changchun, China. [6]Key Laboratory of Zoonosis of Ministry of Agriculture and Rural Affairs, South China Agricultural University, Guangzhou, China. [7]State Key Laboratory of Pathogen and Biosecurity, Beijing Institute of Microbiology and Epidemiology, Beijing, China. [8]CAS Key Laboratory of Pathogenic Microbiology and Immunology, Institute of Microbiology, Chinese Academy of Sciences, Beijing, China. [9]Savaid Medical School, University of Chinese Academy of Sciences, Beijing, China. [10]Guangdong Key Laboratory of Animal Conservation and Resource Utilization, Institute of Zoology, Guangdong Academy of Sciences, Foshan University, Foshan, China. [11]These authors contributed equally: Shouwen Du, Ruchao Peng, Wang Xu, Xiaoyun Qu. ✉e-mail: lwbime@163.com; jxqi@im.ac.cn; liuquan1973@hotmail.com; mliao@scau.edu.cn; lichang78@163.com

pathogens requiring urgent attention by the World Health Organization[8].

SFTSV was initially classified in the *Phlebovirus* genus[1], together with Rift valley fever virus (RVFV), another severe zoonotic virus transmitted by mosquitos. It was later officially named *Dabie banda-virus*, reclassified in the genus *Bandavirus* of *Phenuiviridae* family, order *Bunyavirales*[9]. While the new nomenclature is officially accepted, the term SFTSV has been more widely used in the field. As an enveloped virus, SFTSV invades host cells mainly through the endosomal pathway[10]. The viral genome encodes a membrane protein precursor that undergoes proteolytic maturation to produce two glycoproteins Gn (the N-terminal half) and Gc (the C-terminal half) embedded in the viral envelope[11]. It has been shown that Gn plays a major role in binding the host receptors[12,13], which induces the internalization of viral particles into the endosome[14]. The acidification of endosome further triggers conformational changes of Gc to enable the fusion of endosomal membrane with the viral envelope, allowing the viral genome to enter the cytosol[10].

Previous electron microscopy studies have revealed the icosahedral glycoprotein assemblies, at nanometer resolution, of RVFV[15–17] and Uukuniemi virus (UUKV)[18], two members of *Phlebovirus* genus, suggesting that SFTSV may adopt a similar overall architecture. However, all these viruses have been refractory to high-resolution structure determination due to the high flexibility and deformation of viral particles. By recombinant expression approach, the structures of Gn N-terminal two thirds (Gn head) and the entire ectodomain of Gc from several bunyaviruses have been resolved by X-ray crystallography[19–23]. The SFTSV Gn head contains three domains and was suggested to be responsible for binding host receptors[13,20], in which a few effective neutralizing epitopes have been identified[24–26]. SFTSV Gc displays a typical fold of class II viral fusion proteins and was only crystalized as a

homotrimer in the postfusion conformation[21]. Interestingly, an earlier study reported a structure of RVFV Gc crystalized as a head-to-tail homodimer similar to flavivirus E proteins in the pre-fusion conformation[22]. In addition, recombinant expression of the full-length ectodomain of SFTSV and RVFV Gn proteins produces disulfide linked homodimers[20], which makes the organization of Gn and Gc on virion surface even more promiscuous. This knowledge gap greatly hindered our understanding of the mechanisms of bunyavirus assembly and infecting host cells.

In this work, we report the near-atomic resolution structure of SFTSV virion determined by cryogenic electron microscopy (cryo-EM) single particle analysis and sub-particle local reconstruction. This structure illuminates the molecular basis underpinning the assembly of Gn and Gc glycoproteins on virion surface, and suggests the mechanisms of membrane fusion mediated by conformational changes of the glycoproteins, as well as viral infection inhibition by neutralizing antibodies.

## Results

### Structure determination and overall architecture of SFTSV virion

To characterize the structure of SFTSV virion, we first performed cryogenic electron tomography (cryo-ET) analysis on purified virions. The tomograms revealed approximately spherical viral particles with some local distortions, suggesting the flexibility of the virion (Fig. 1a). Slicing through the tomograms, we observed a clear lattice consisting of hexagonal and pentagonal densities on the viral envelope (Fig. 1b), indicating the approximately icosahedral symmetry of the virion. Since there are more hexagons than pentagons in each virion, individual hexagonal particles were extracted, aligned, and classified to improve the resolution, by a hybrid subtomogram averaging and single particle

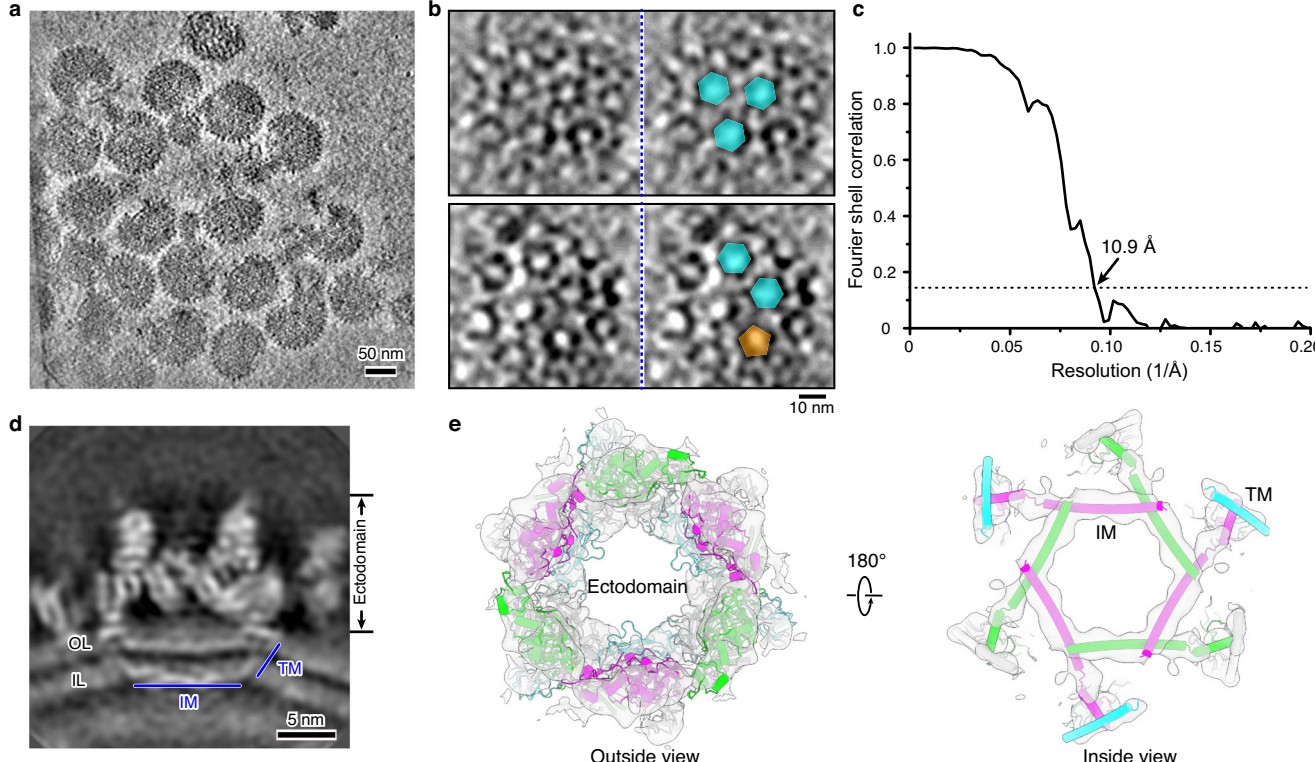

**Fig. 1 | Cryo-ET analysis of SFTSV virion. a** A representative tomogram slice of SFTSV virions. A total of 118 tomograms were collected for this study. **b** Zoom-in tomogram slices showing the pentagonal (orange) and hexagonal (cyan) densities on virion surface. The left and right panels are essentially the same except that the latter are annotated with pentagons and hexagons. **c** FSC curve for the subtomogram averaging density map of the hexon peplomer. **d** Cross-section view of the hexon density map produced by subtomogram averaging. OL outer leaflet, IL Inner leaflet, TM transmembrane domain, IM intramembrane domain. **e** The density map of hexon peplomer fitted with a pseudo-atomic model. The map is shown as transparent surface to reveal the tubular feature overlaid with helices.

reconstruction workflow[27], which produced a density map at 10.9 Å resolution (Fig. 1c). This structure reveals clear tubular densities of α-helices that span across the viral envelope or are embedded in the inner leaflet of the membrane (Fig. 1d, e). To achieve the best resolution possible, we turned to single particle analysis method for structure determination (Supplementary Fig. 1). Global three-dimensional (3D) reconstruction applying icosahedral symmetry resulted in a density map at 6.7 Å resolution (Supplementary Fig. 1c). To counteract the defocus gradient and local structural heterogeneity within the virion, block-based reconstruction[28] was performed for sub-particles at 2-fold, 3-fold and 5-fold symmetry axes, which improved the resolution of these blocks to 4.8 Å, 4.8 Å, and 4.6 Å, respectively, with the local resolution of the central regions reaching ~4.0 Å (Supplementary Figs. 1c, 2a). These density maps allow unambiguous identification of Gn and Gc glycoproteins that exist as heterodimers on the viral envelope. The density of Gn/Gc heterodimers were further averaged to facilitate atomic modeling, which clearly resolved the main chain trace of each protein and some bulky side chains could also be recognized (Supplementary Figs. 1–3).

The SFTSV virion is about 110 nm in diameter, arranged approximately in icosahedral symmetry with a triangulation (T) number of 12 (Fig. 2), similar to RVFV[15–17] and UUKV[18]. The ectodomains of Gn and Gc tightly pack together as a heterodimer, which further assembles into pentameric (penton) and hexameric (hexon) peplomers projecting from the viral envelope (Fig. 2e). In each virion, there are 12 pentons sitting at the icosahedral vertices, and 110 hexons covering the facets, giving a total of 720 Gn/Gc heterodimers in each virion. According to the topology within the icosahedron, the hexons can be divided into three types, peripentonal (P), edge (E), and center (C), which are quasi-equivalent in structure (Fig. 2c). In both pentons and hexons, the Gn heads cluster together to form a crown on top of Gc subunits, making Gc less accessible to the solvent (Fig. 2e). This is consistent with the observation that many neutralizing antibodies inhibit virus infection by targeting the Gn head[26,29], which also suggests it is an ideal candidate for developing subunit vaccines. Inside the viral envelope, a hexagonal network of helices is observed at the bottom of hexons (Fig. 2b), assembled by the C-terminal intramembrane (IM) helix of Gn following the transmembrane (TM) domain. In contrast, this region

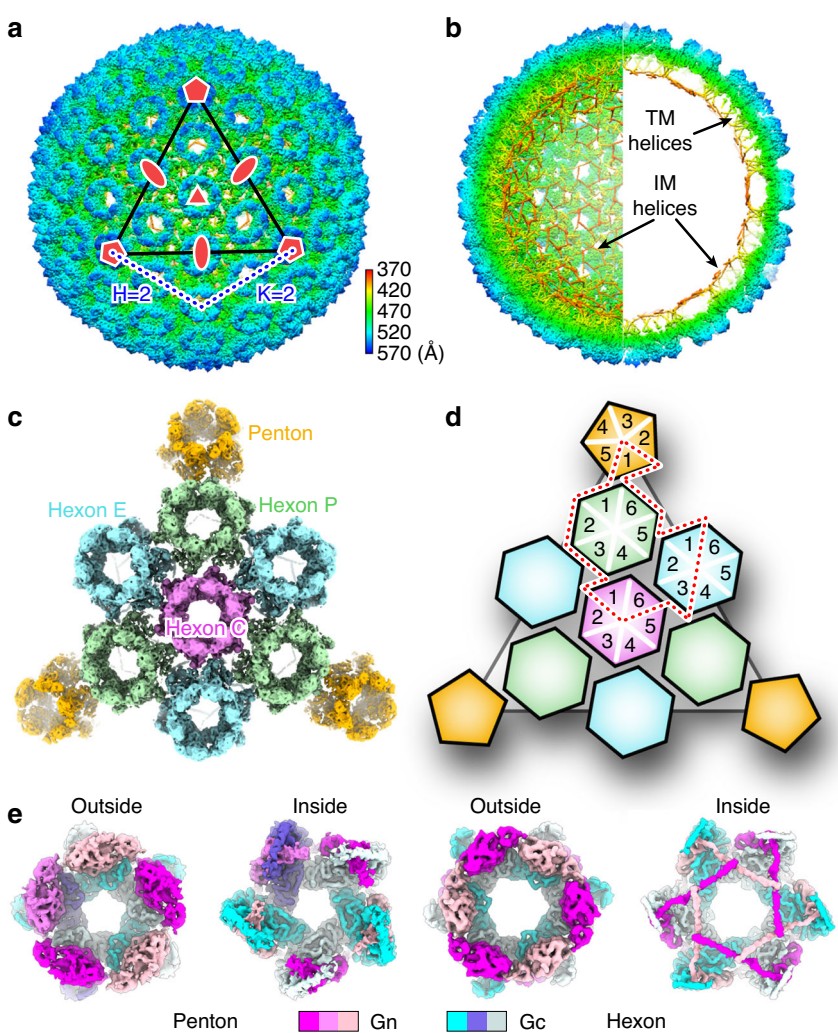

**Fig. 2 | Overall structure of SFTSV virion. a** Cryo-EM density map of SFTSV virion reconstructed with icosahedral symmetry, colored by the radial distance from the virion center. One facet of the icosahedron is indicated by a black triangle. The 5-, 3- and 2-fold symmetry axes are represented by pentagon, triangle, and oval, respectively. Triangulation information of the virion is indicated by dashed lines, where T = H² + K² + HK equals 12. **b** Cross-section view of the density map to visualize the intramembrane (IM) helix lattice and transmembrane (TM) regions, indicated by arrows. **c** Density map of a facet within the icosahedral virion, colored by different types of peplomers. There are three types of hexons, peripentonal (P), edge (E) and center (C). **d** Schematic model of the organization of different peplomers, colored in the same code as in **c**. One ASU is outlined with dashed lines. **e** Architecture of penton and hexon peplomers viewed from outside or inside of the virion. The density is colored by different subunits as depicted in the legend below.

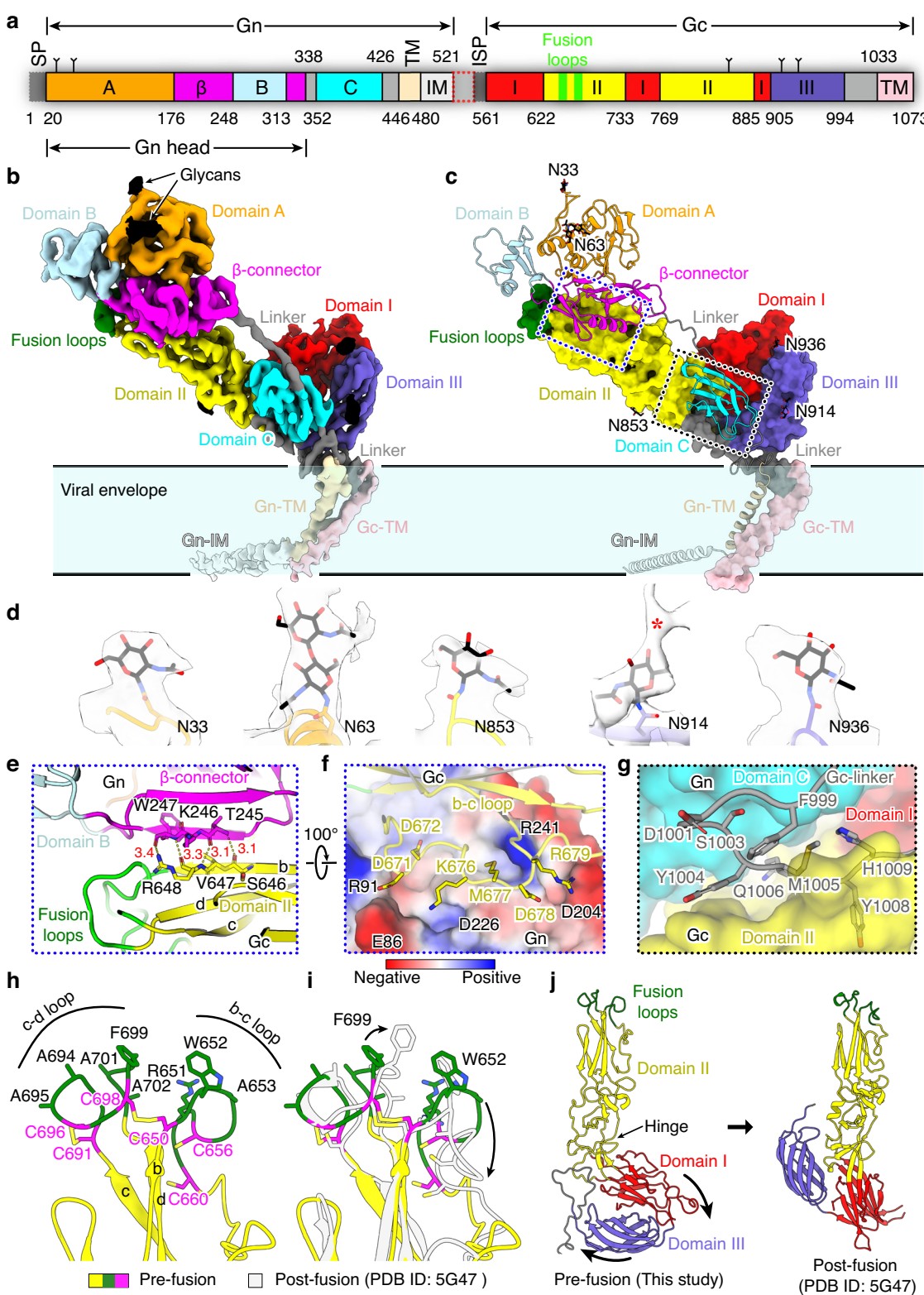

was not resolved in pentons, indicating the flexible conformation in pentameric context (Fig. 2e).

**Structure of Gn/Gc heterodimer**

The ectodomains of Gn and Gc both display an elongated shape and pack in a parallel orientation, of which the Gn head is slightly twisted to dock on top of Gc domain II (Fig. 3). The Gn head consists of three domains (A, β-connector and B). Domain A is fully exposed on the top

and contains two N-linked glycans on residues N33 and N63 (Fig. 3a–d). It was reported that SFTSV can utilize DC-SIGN, a glycan binding protein, as the receptor to enter host cells[12,14], suggesting a role of these glycans in virus entry. The β-connector domain of Gn constitutes the major interface to interact with Gc within the heterodimer, and domain B forms an appendage to cap the fusion loops at the tip of Gc domain II (Fig. 3b, c). Previous studies have reported the structures of Gn head from a few phenuiviruses and hantaviruses[19,20]. Even though the

**Fig. 3 | Structure of SFTSV Gn/Gc heterodimer. a** Schematic diagram of the domain architecture of Gn and Gc in the context of the glycoprotein precursor. Each domain is represented by a unique color, and the glycosylation sites are indicated by small branches. The signal peptide (SP) of Gn and the internal signal peptide (ISP) of Gc are colored in dark gray, and the linker between Gn and Gc is outlined with red dashed lines. **b, c** The cryo-EM density map **b** and atomic model **c** of SFTSV Gn/Gc heterodimer colored by domains, with the same code as in **a**. The N-linked glycans are colored in black, and the fusion loops are highlighted in green. The blue and black dashed boxes indicate the key interaction interfaces between Gn and Gc, for which the details are shown in **e**–**g**. **d** Density map of the glycans in close-up views. The red asterisk indicates the unmodeled density for additional glycan residues. **e**–**g** Molecular interactions at the Gn/Gc interface. The structures are shown in cartoons and colored by domains as in **a**. The residues potentially involved in Gn/Gc interactions are shown as sticks and colored by elements. In **e**, the dashed lines represent the hydrogen bonds between Gc b-strand and the adjacent Gn strand in the β-connector domain. The distances between hydrogen donor and acceptor atoms are labeled with red numbers (in Å). In **f**, the electrostatic potential surface of Gn is shown to depict the complementary charge pattern with Gc residues. In **g**, the Gn domain C and Gc domain I and domain II are shown in surface, and the long Gc linker is presented with ribbon model which inserts into the crevice between Gn domain C and Gc domain II. **h** Structure of the fusion loops in SFTSV Gc. The structure is colored by domains and the fusion loop regions are colored in green. The key hydrophobic residues (green) and cysteines (magenta) are shown as sticks. **i** Superposition of Gc structures before (colored by domains) and after (white) membrane fusion in the fusion loop region. The conformational changes of the fusion loops and key residues are indicated by arrows. **j** Comparison of SFTSV Gc structures in the pre- (left) and post-fusion (right) conformations. The hinge region between domain II and domain I is highlighted with an arrow, which becomes extended in the postfusion conformation. The directions for domain movement are indicated by curved arrows.

structural fold of these proteins varies, the overall architecture of the three domains is quite similar (Supplementary Fig. 4). Particularly, the functional equivalent of phenuivirus Gn domain B in hantavirus is a long β-hairpin in domain A which projects a loop (termed the cap loop) to cap the Gc fusion loops in a similar topology[19] (Supplementary Fig. 4c, d). The Gn domain C displays an Ig-like fold and binds to Gc at the hinge region between domain I and domain II, which facilitates stabilizing the metastable prefusion conformation of Gc (Fig. 3b, c). Between the Gn head and domain C is a long flexible linker that presumably allows Gn to undergo dramatic conformational changes to expose Gc for membrane fusion (Fig. 3b, c).

The structure of Gc in the prefusion conformation adopts an arch configuration in which domain I folds towards domain II, creating a valley at the hinge region. In addition, domain III tightly packs against domain I to make the structure in a compressed conformation (Fig. 3c). This is in sharp contrast to the "relaxed" postfusion conformation in which the hinge between domain II and domain I is extended and domain III binds laterally to this region[22,30] (Fig. 3j). The energetically unfavorable prefusion conformation of Gc is essentially maintained by interactions with Gn as well as further assembly of high-order peplomers, which may explain the failure of crystalizing Gc in the prefusion conformation (Supplementary Fig. 5). There are three N-linked glycosylation sites in SFTSV Gc, of which N914 is highly conserved among all phleboviruses and some bandaviruses (Fig. 3a–d; Supplementary Fig. 6).

The interactions between Gn and Gc mainly involve two discrete interfaces, mediated by the Gn head and domain C, respectively (Fig. 3c). At the top region, two antiparallel strands in Gn β-connector domain pair with the b-c-d strands within Gc domain II, resulting in an extended five-strand β-sheet, in which residues T245-W247 of Gn and S646-R648 of Gc form a hydrogen bond network involving both the main chain and the side chain (Fig. 3e). Below the sheet, the b-c loop of Gc domain II forms extensive polar contacts and electrostatic interactions with the bottom of Gn β-connector domain (Fig. 3f). In the lower part, the Gn domain C adheres to the junction region between Gc domain I and domain II, and the long membrane-proximal linker loop of Gc inserts into the cleft to further secure the interactions (Fig. 3g). To initiate membrane fusion, Gn must dissociate from Gc to unlock its conformation and allow membrane targeting by the fusion loops.

Unlike the single fusion loop of flavivirus and alphavirus E proteins[31,32], the membrane targeting motifs of SFTSV Gc include two loops (b-c and c-d loops) stabilized by two sets of conserved disulfide bonds (Fig. 3e, h). Besides, a pair of free cysteines (C656 and C660) is present in the b-c loop, which allows it to flip up and down before and after membrane fusion (Fig. 3h, i). The conserved aromatic residue F699 is erected in the postfusion conformation in contrast to the lying conformation at the prefusion state, suggesting a more favorable conformation for membrane targeting (Fig. 3i). In addition, residue W652 slightly moves downwards, together with the entire b-c loop, in

the post-fusion conformation, possibly a result of concerted conformational change to create space for accommodating the erected c-d loop (Fig. 3i).

The role of histidine serving as a pH sensor in viral fusion proteins has been well documented[33]. A few histidine clusters are also observed in SFTSV Gn/Gc heterodimer that are potentially the key switches regulating the conformation of Gc under different pH conditions (Supplementary Fig. 6a–d). In the domain C of Gn, residue H407 is surrounded by K623 and R624 in Gc domain II; residues K409, S372, and T374 of Gn may potentially interact with Gc residue H870 (Supplementary Fig. 6b). In neutral pH, histidines can interact with these residues via polar contacts. Once protonated under acidic conditions, the positive charge will force histidines to repel other basic residues, leading to deformation of the interface. Besides, H396 in Gn domain C clusters with H1009 in the membrane proximal linker of Gc, and residues H568, H606, and H940 of Gc form a triad at the hinge between domain I and domain III (Supplementary Fig. 6c, d). All these patches may contribute to the conformational changes of Gc in the endosome to mediate membrane fusion, among which H407 of Gn is highly conserved for all phleboviruses and some bandaviruses (Supplementary Fig. 6f).

## Assembly of hexon and penton peplomers

At the virion surface, Gn/Gc heterodimers, as the basic structural unit, intercalate with each other via lateral interactions to assemble into penton and hexon peplomers (Fig. 4). Each peplomer displays a three-layered architecture, a crown formed by the Gn head, the trunk comprised by Gc subunits and Gn domain C, and the TM regions (Fig. 4a, h). In the crown of hexons (with hexon C as an example), an edge loop in domain B protrudes into a crevice in domain A of the adjacent Gn subunit, producing a closed ring structure (Fig. 4b). This interface contains multiple charged residues and is possibly stabilized by complementary electrostatic interactions (Fig. 4c). At the bottom of each hexon, the IM helix of Gn bridges its neighboring counterpart in a head-to-tail orientation, creating a locked hexagonal network in the inner leaflet of membrane (Figs. 2b and 4d). Within the trunk region, the valley between domain I and domain II of each Gc subunit perfectly accommodates the domain II from the adjacent subunit (Fig. 4e). In this context, the fusion loops of Gc are sandwiched by domain B of the Gn head and domain II from a neighboring Gc subunit, where a cap helix in Gn domain B covers the top and the g-h loop of Gc domain II supports the bottom (Fig. 4f). This intertwined configuration ensures the hydrophobic fusion loops are well protected from exposure to solvent, which in turn requires the peplomer to dissociate in order to allow membrane targeting by the fusion loops. Besides, a stem helix (α4) in Gn β-connector domain docks on top of Gc domain I in the neighboring heterodimer, which further secures the interactions within each peplomer (Fig. 4e). This helix contains a few negatively charged or polar residues, so that nicely binding to the highly

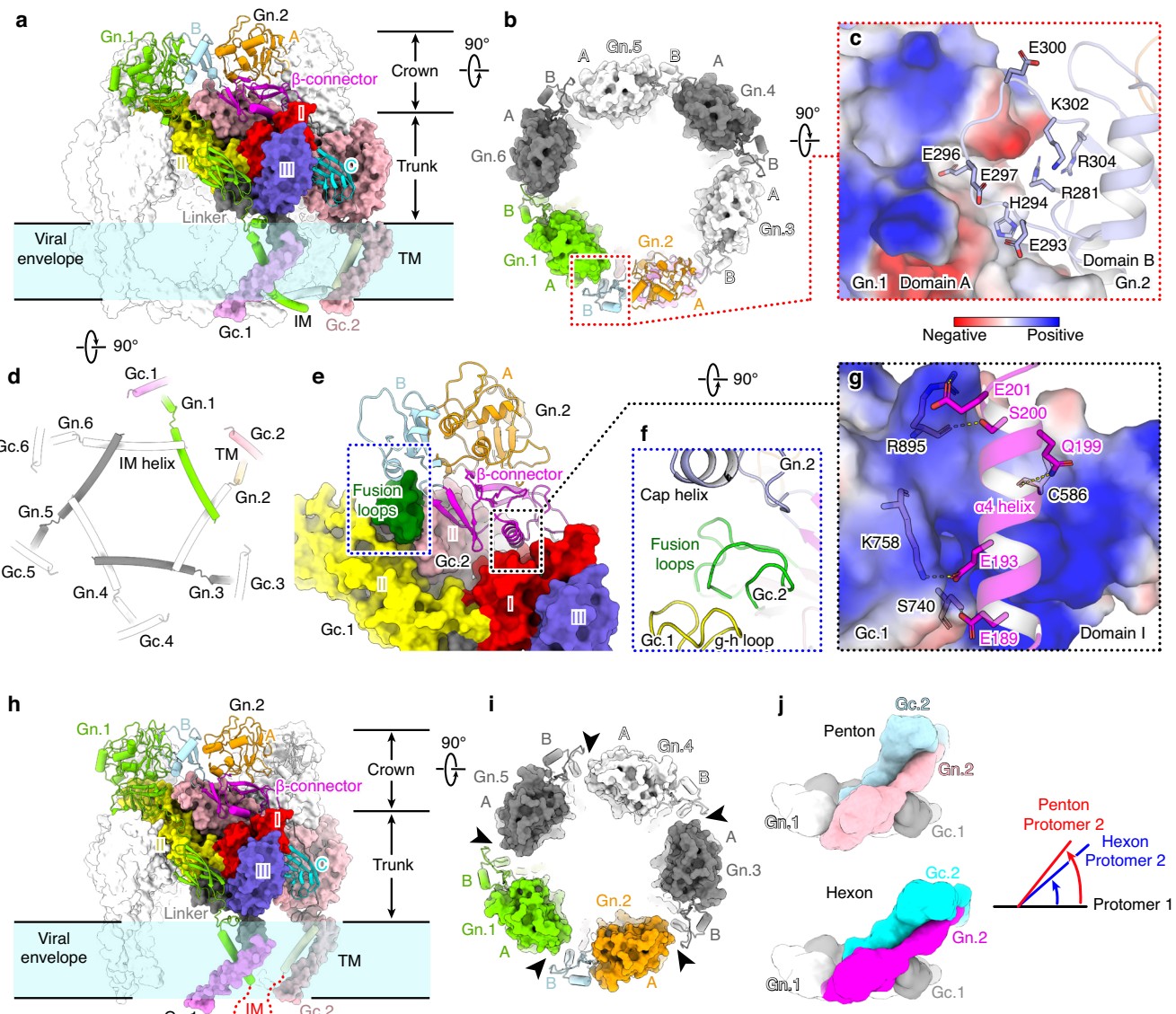

**Fig. 4 | Structural basis of SFTSV peplomer assembly. a** Structure of the hexon peplomer (depicted with hexon C structure). For clarity of visualizing inter-heterodimer interactions, two adjacent Gn/Gc heterodimers are colored by domains or subunits, with the other subunits colored in white and set to transparent. **b** Top view of the hexon to visualize the interactions in the crown region. Domain B of each subunit is shown in cartoons to reveal its interactions with adjacent subunits. **c** Zoom-in view of the interactions between Gn domain B and the domain A from an adjacent subunit. **d** Bottom view of the IM helix network in hexons. **e** The interface between two adjacent Gn/Gc heterodimers. The Gn subunit is shown in cartoons and Gc subunits in surface models, colored by domains. The fusion loops are highlighted in green. **f** Close-up view of the molecular context shielding the Gc fusion loops. The fusion loops are sandwiched by the Gn domain B and a neighboring Gc subunit. **g** Complementary electrostatic interactions between Gn α4 helix and Gc domain I from the neighboring heterodimer. The key hydrogen bonds and salt bridges are shown as dashed lines. **h–i** Structure of the penton peplomer in side **h** and top **i** views, shown in a similar manner as the hexon in **a**, **b**. **j** Comparison of the relative orientations between two adjacent Gn/Gc heterodimers in pentons and hexons.

positively charged surface of Gc domain I. Residues E201 and E193 of Gn may potentially form two salt bridges with R895 and K785 in Gc, respectively, and residues E189, Q199 and S200 of Gn likely interact with S740, C586 and R895 of Gc via hydrogen bonds (Fig. 4g). The interaction pattern within pentons is similar to that in hexons except that the relative orientation of two adjacent Gn/Gc heterodimers is a bit different due to distinct symmetries, which leads to slightly different atomic interactions at the interface (Fig. 4h–j). Importantly, the crown of pentons is not locked as the Gn domain B is farther away from the neighboring domain A than in hexons, engendering a gap between two adjacent Gn subunits (Fig. 4i). In addition, the IM helix of Gn is disordered in pentons (Fig. 4h). These differences suggest the structure of SFTSV pentons might be less stable than hexons.

Because of the different molecular contexts, the conformations of the three types of hexon exhibit some local variations (Supplementary Fig. 7a). Superposition of the 12 Gn/Gc heterodimers within an asymmetric unit (ASU) shows that the conformations of the ectodomains are quite consistent while the TM regions display noticeably different orientations (Supplementary Fig. 7b). This might be attributable to the long membrane proximal linkers of Gn and Gc which allow the domains connected at the two ends to adopt different conformations (Supplementary Figs. 4b, 5b). Structural clustering of the 12 quasi-equivalent Gn/Gc conformers revealed three clusters, with the 11 hexonal heterodimers clustering into two closely related groups, and the one in pentons being most distant from the others (Supplementary Fig. 7c).

## Interactions between individual peplomers

The assembly of penton and hexon peplomers are mainly mediated by interactions in the ectodomains which form an ordered contacting network to stabilize the icosahedral glycoprotein shell (Fig. 5a; Supplementary Fig. 8). Inside the envelope, the TM helix of Gc also contributes to inter-peplomer interactions, which plays a much minor role (Fig. 5a). Across the entire glycoprotein lattice, there are four types of inter-peplomer interactions underpinning the assembly of SFTSV viral particle. Type 1 interaction occurs at the interface between the penton and the P hexon, as well as neighboring hexons, with two sets of interactions between two neighboring peplomers (Fig. 5a). Exemplified by the penton-hexon interface, the outer sheet of Gc domain III in the penton adheres to the end loops of the Ig-like domain C of hexon Gn, which is the most prevalent interactions between individual peplomers. Three sets of such interactions interlock three adjacent peplomers, creating a closed interaction network between a penton and two P hexons, or within three neighboring hexons (Supplementary Fig. 8b–d). This interface involves both polar and non-polar residues, and the highly conserved glycans on residue N914 may also contribute to the interactions (Fig. 5b). Although only a single sugar moiety was modeled on N914, some additional density was observed that likely represented a polysaccharide chain but was too weak to be confidently modeled, possibly due to the heterogeneous conformations of the glycans (Fig. 3d). To test the functional importance of the glycans, we generated an N914Q mutant that abolished the glycosylation on this residue. Despite the mutant protein was expressed at a similar level to the WT in cells, the production of Gn/Gc-pseudotyped viral particles was severely impaired, suggesting the crucial role of the glycans for SFTSV virion assembly (Fig. 5f, g).

Type 2 interaction exists at the interfaces between penton and P hexon, C and E hexons, as well as neighboring P hexons (Fig. 5a; Supplementary Fig. 8h–j). This interface is formed mainly by two Gc domain III arranged in a head-to-head orientation, resulting in a symmetric interaction pattern, which might be stabilized by electrostatic interactions (Fig. 5c). Type 3 interaction only occurs between the penton and two adjacent P hexons in the TM region (Supplementary Fig. 8e). The C-terminal end of TM helix from the two hexon Gc are in close proximity, potentially interacting through polar contacts. The TM helix of penton Gc is a bit farther away from the two hexon Gc, with a distance of more than 10 Å between the closest Cα atoms, possibly not contributing to the interactions (Fig. 5d). Type 4 interaction is a slightly different scenario, which occurs at the interface of three hexons (Supplementary Fig. 8f, g). The three Gc TM helices are roughly symmetrically arranged, creating an intercalated network possibly stabilized by polar interactions (Fig. 5e).

## Comparison of peplomer topology in related class II enveloped viruses

Shielding the fusion peptide is well accepted as a general theme for enveloped viruses to maintain the prefusion conformation of fusogenic glycoproteins, which could be achieved via different mechanisms. Among them, a few different viruses are known to utilize class II fusion proteins and shield the fusion loop(s) by an accompanying glycoprotein, e.g., alphaviruses[34,35], rubiviruses[36], phenuiviruses[16] and hantaviruses[19]. Though the virion structure and the fold of individual glycoproteins significantly differ among these viruses, the overall architecture of the glycoprotein heterodimers are remarkably conserved (Fig. 6). The fusion loop(s) are capped by different domains of the accompanying protein, all of which form an appendage at one end of the heterodimer with a similar geometry relative to the fusion peptide (Fig. 6e–g). Nevertheless, the capping domain in different viruses shows distinct topologies relative to other domains. In alphaviruses, the capping domain (B) is linked to the β-ribbon connector, similar to phenuiviruses, though the β-connector of the latter is a more elaborate domain containing more secondary structural elements.

For hantaviruses, however, the β-ribbon connector is more integrated as a part of the large domain B, and the capping element is a long hairpin loop that resides within domain A (Fig. 6e–g). These observations highlight the structural diversity of class II viral glycoproteins that serve similar functions for viral assembly and cell entry.

Within the penton and hexon peplomers, the fusion loops of phenuiviruses are further buried by neighboring Gn/Gc heterodimers. In comparison, the fusion loop(s) of hantaviruses and alphaviruses are relatively more accessible in the context of tetrameric or trimeric spikes, respectively (Fig. 6a–d). Besides, the domain C of the accompanying glycoprotein of alphaviruses (E2) and hantaviruses (Gn) is wrapped inside the spikes, which glues the adjacent heterodimers to stabilize the trimeric/tetrameric spikes (Fig. 6c, d), while the inter-spike interactions are mainly mediated by the fusogenic proteins[19,37]. For SFTSV, however, the Gn domain C is not involved in interactions between adjacent Gn/Gc heterodimers within pentons/hexons, which is exposed in the periphery of each peplomer and contributes to inter-peplomer interactions together with the fusogenic Gc (Figs. 5b and 6a, b). The diverse assembly of glycoprotein spikes/peplomers might be specialized in favor of receptor binding or membrane fusion of different viruses. A few studies have suggested that SFTSV bind receptors via the Gn head[13,14,25], which is fully exposed at the top of peplomers. By contrast, chikungunya virus, an alphavirus, utilizes both E1 and E2 proteins to recognize its receptor which binds in the crevice within the trimeric spike[34,35].

## Discussion

Although SFTSV virions, as well as other phenuiviruses, display an icosahedral architecture, the conformation is relatively more flexible than other large icosahedral viral capsids, such as herpesvirus and adenovirus. This is probably because the inter-peplomer interactions only involve a small portion of the glycoprotein surface, and the TM regions do not contribute much to the interaction. In addition, phenuiviruses lack various "minor capsid proteins" as found in herpesviruses that cement the inter-capsomer/peplomer spaces to further secure the protein shell[38]. These features render the phenuivirus virion prone to deform locally, resulting in partially pleomorphic virion structures. This flexibility may in turn facilitate the conformational changes of glycoproteins for membrane fusion, which would otherwise be unfavorable for virus entry as the fusogenic Gc is quite stably buried in each peplomer.

One remarkable feature of SFTSV glycoproteins is that both Gn and Gc have a long membrane-proximal linker loop connecting the ectodomain and the TM helix (Supplementary Figs. 4b, 5b). This linker allows the Gn/Gc ectodomains to adopt different orientations relative to the viral envelope. In the context of penton and hexon peplomers, the Gn/Gc ectodomains are tilted from the viral membrane with a shallow angle, making the Gc fusion loops distant from the target membrane. Once activated by the low pH environment, the Gn/Gc heterodimer may tilt to a higher angle to place the fusion loops in close proximity to the endosomal membrane. Moreover, the dislocation of Gn allows the rearrangement of Gc domains to transform into an extended conformation, enabling the insertion of fusion loops into the endosomal membrane, followed by further conformational changes to trigger membrane fusion (Supplementary Fig. 9). A few Gc structures in the extended conformation have been observed for both phenuiviruses and hantaviruses[22,30], which may represent an intermediate state competent for membrane targeting (Supplementary Fig. 5c, d).

The high-order assembly of peplomers stabilizes the virion structure at the prefusion state, and also creates some cryptic epitopes that are not accessible in the mature virion before entering the endosome. Only if the viral particles undergo conformational changes will these epitopes become exposed for antibody binding.

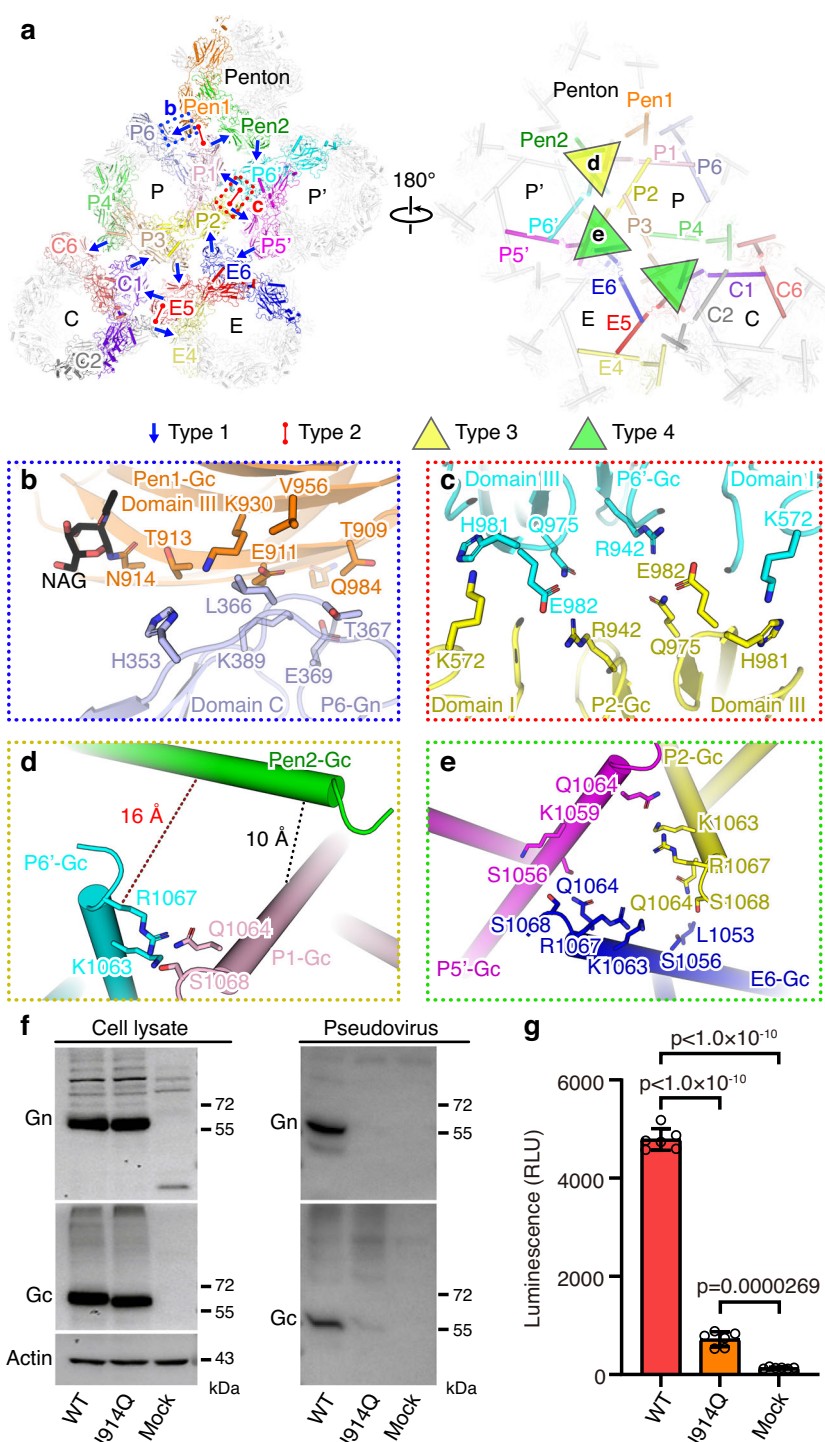

**Fig. 5 | Inter-peplomer interactions underpinning the assembly of SFTSV virions. a** Schematic depiction of the inter-peplomer interactions within an ASU. Each Gn/Gc heterodimer at the interface is represented with a unique color. The four types of interactions are labeled on the structure as the legend below. The blue and red dashed boxes are selected examples to analyze atomic details for type 1 and 2 inter-peplomer interactions shown in **b** and **c**, respectively. The selected examples of type 3 and 4 interactions are labeled with **d**, **e**, respectively. **b**–**e** Close-up view of the four types of molecular interactions at the interface between adjacent peplomers. Each type of interaction is outlined with a dashed box in the same color code as in **a**. The structures are shown in cartoons and colored by molecules as in **a**. The potential interacting residues at the interface are shown as sticks and colored by elements. In panel **d**, the nearest distances between Cα atoms in the neighboring

helices are shown as dashed lines. **f** Western blot analysis of WT and mutant glycoprotein expressions in cells, as well as the comparison of Gn/Gc in WT and mutant pseudoviruses. The data shown is a representative result of 3 independent experiments using different sample preparations. The uncropped blots are provided in the Source Data. **g** Comparison of infectivity of WT or mutant Gn/Gc pseudo-typed viruses. The data shown is a representative result of 3 independent replicative experiments using different sample preparations, presented as the mean values (histogram) and standard deviations (error bar) of $n = 6$ replicas (dots) in one experiment. Source data is provided for the plot. The significance of the difference was tested by one-way ANOVA. *P* values are shown on the histogram. RLU, relative light unit.

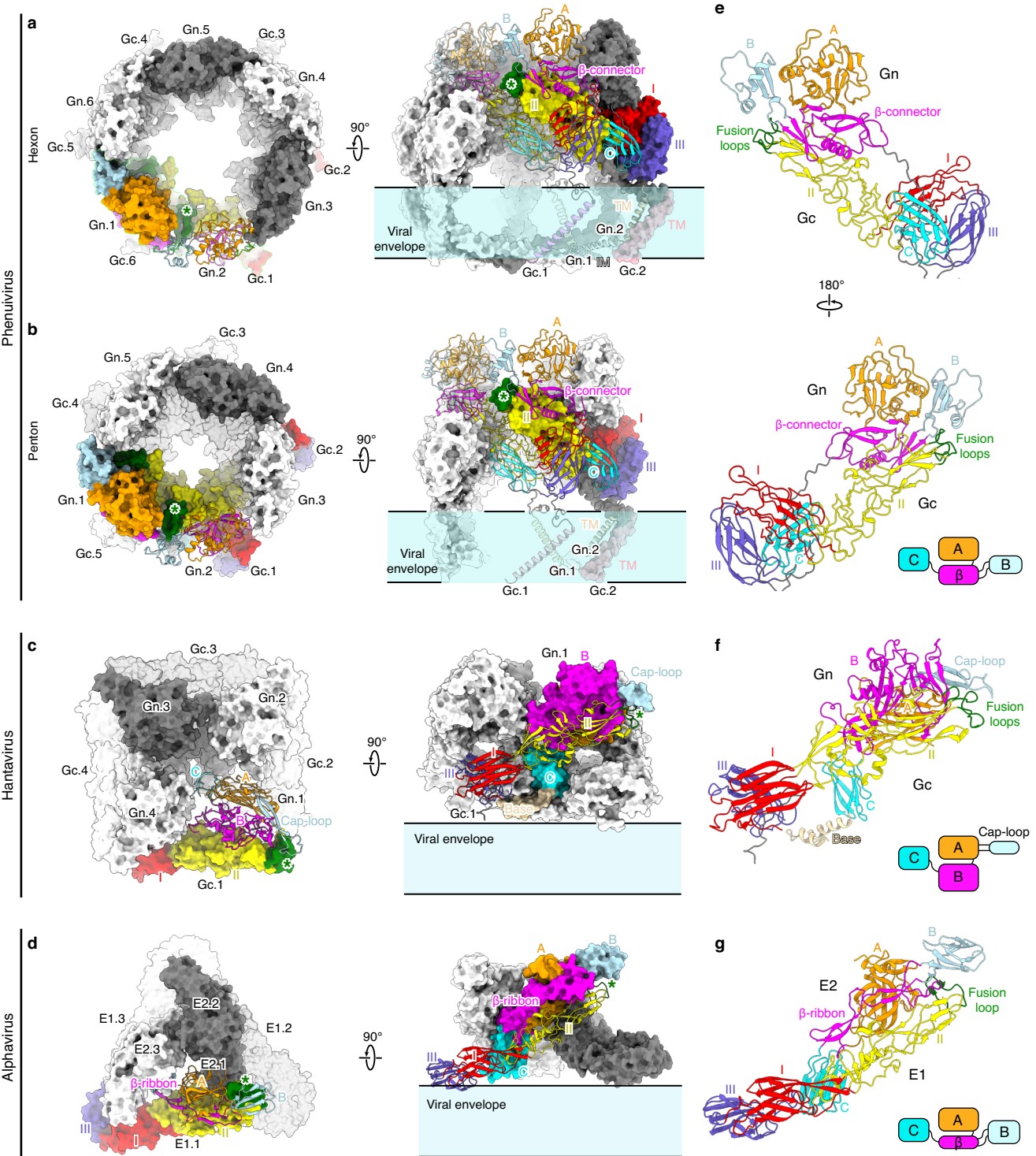

**Fig. 6 | Comparison of different peplomer assemblies in bunyaviruses and alphaviruses. a–d** Phenuiviruses, represented by SFTSV, are icosahedral with hexon **a** and penton **b** peplomers, while hantaviruses, represented by Andes virus[19] (PDB ID: 6ZJM), are pleomorphic with square-shaped tetrameric spikes/peplomers **c**. Alphaviruses, represented by Chikungunya virus[34] (PDB ID: 6JO8), are also icosahedral but with trimeric spikes/peplomers **d**. All these peplomers are assembled with a heterodimeric glycoprotein complex as the structural unit. For clarity, one heterodimer in each peplomer is colored by domains and the other copies are colored in gray or white accordingly. The fusion loop(s) are highlighted by green asterisks. Both top and side views of the peplomers are shown to better reveal the structural domains shielding the fusion loop(s). **e–g** Structural comparison of the heterodimeric glycoproteins from related class II viruses. The structures are shown in cartoons and colored by domains as in **a–d**. The bottom-right inset in each panel shows the domain topology diagram of the accompanying glycoprotein for each virus. The functional equivalent domains are represented with the same color.

A substantial number of such epitopes have been identified in SFTSV Gn head[24–26], in addition to those fully exposed at the top of domain A that potentially interfere with receptor binding (Supplementary Fig. 10a). This indicates that many antibodies neutralize viral infection by targeting the transient intermediate states in the process of membrane fusion. Among them, the monoclonal antibody MAb4-5, targeting the cap helix in Gn domain B[20], displays effective neutralizing activities against a broad range of SFTSV isolates[25], for which the

mechanism of viral neutralization is not yet well understood. In the prefusion state of mature virion, this epitope is shielded by the adjacent Gn/Gc subunits in the peplomer, which renders steric hindrance for antibody binding (Supplementary Fig. 10a). However, it was shown that MAb4-5 can bind to SFTSV viral particles when incubated at room temperature in neutral conditions[25], implying "breathing" of virions that transiently exposes the epitope as observed in flaviviruses[39]. The frequency of such stochastic conformational switch is pretty low, whereas it would be much more robust when exposed to the acidic environment in the endosome, indicating the binding of MAb4-5 to SFTSV virions may mainly occur in the endosome. This is similar to the scenarios of many fusion loop targeting antibodies of flaviviruses which may largely require the antibodies to be internalized into the endosome together with the viral particles[40–42]. However, it is yet unclear how these antibodies ensure efficient uptake by the endosome to enable viral neutralization in the membrane fusion process. Docking the structure of MAb4-5 onto SFTSV Gn/Gc prefusion heterodimer, by aligning the Gn head, revealed that the complementarity-determining region 2 (CDR2) of the heavy chain is in close proximity to the fusion loops, potentially interacting with the highly conserved aromatic residue F699 (Supplementary Fig. 10b, c). The binding of MAb4-5 thus creates an additional shield for the fusion loops to prevent their conformational changes for membrane targeting. A previous study showed an intermediate state of RVFV binding to a liposome in which the fusion loops of Gc were inserted into the membrane and the Gn head was slightly retracted and bound to the lower part of Gc domain II[43]. This structure is in line with our hypothesis that MAb4-5 locks an earlier conformation of Gn/Gc heterodimer to prevent the exposure of fusion loops for membrane targeting (Supplementary Fig. 9b, c).

While the manuscript was under peer-review, Sun et al. reported a similar structure though at slightly lower resolution in some local regions[44]. By similar block-based reconstruction strategy, they resolved the structures of 5-, 3- and 2-fold blocks at 7.2 Å, 5.2 Å, and 5.9 Å resolutions, respectively. Moreover, the 3- and 2-fold blocks were further processed by in situ single particle analysis (isSPA) approach[45], which improved the resolutions to 4.5 Å and 5.2 Å, respectively. These maps revealed the global icosahedral architecture of the peplomers on virion surface and allowed the authors to propose the model of the Gn/Gc glycoproteins, leading to similar findings of our study. However, they were unable to resolve the IM helix of Gn in hexons as well as some glycans in Gc, including the critical N914 glycans that contribute to virion assembly. In addition, the resolution of pentons was limited to the secondary structure level in their structure. These differences might be largely attributable to the extreme flexibility of phenuiviruses, which leads to extensive local conformational heterogeneity across the virion. The slightly better resolution of our structure, as well as being able to resolve these additional features, might be due to the larger size of our data set that allowed us to isolate relatively homogeneous subsets with sufficient sub-particles after intensive classification to generate the final reconstructions.

Collectively, our study allows mechanistic insights into bunya-virus assembly in unprecedented near-atomic details, which illuminates the mechanisms of viral infection as well as its inhibition by neutralizing antibodies. These advances would provide an important basis for developing vaccines and antiviral therapeutics.

## Methods

### Cell culture and virus propagation
African green monkey kidney (Vero), HEK293T/17 and HEK293 cells were cultured in Dulbecco's modified Eagle medium (DMEM, Hyclone) containing 10% fetal bovine serum (Gibco), 100 U/mL penicillin and 100 μg/mL streptomycin. All cells were donated by the Stem Cell Bank, Chinese Academy of Sciences. SFTSV strain JA1-2018 (ref. 46), isolated from a tick-bitten patient in Jilin, northeastern China, was propagated

in Vero cells, cultured at 37 °C in the atmosphere with 5% $CO_2$. The supernatant was collected at 72 h post-infection (hpi) and clarified by centrifugation at $1000 \times g$ speed for 10 min at 4 °C. The cleared viral suspension was stored in aliquots at −80 °C before further usage. The virus titer was titrated as 50% tissue culture infectious dose (TCID50) according to the Reed−Muench method[47].

### Virus purification
To purify SFTSV virions for structural analysis, Vero cells grown on 75 cm² side-bottom tissue culture flasks were infected with SFTSV at a multiplicity of infection (MOI) of 2 and cultured at 37 °C in the atmosphere with 5% $CO_2$ for 72 h. The cell culture medium was harvested by centrifugation at $1000 \times g$ speed for 10 min at 4 °C, and cleared through a 0.22-μm cut-off nitrocellulose filter. The virus sample was then fixed with 1% paraformaldehyde (EM grade, EM sciences), for 1 h at 4 °C, and concentrated by ultracentrifugation through a 20% (w/v) sucrose cushion which was centrifuged at $154,300 \times g$ with a Beckman SW41 rotor for 2 h at 4 °C. The pelleted viral particles were then resuspended in 1 × Phosphate Buffered Saline (PBS, pH 7.4) and the concentration was determined by the Bicinchoninic acid (BCA)-based protein quantification method (Yeasen Biotechnology).

### Cryo-EM/ET sample preparation and data collection
A 4-μL aliquot of virus sample (3.6 mg/mL, in 1×PBS) supplemented with 0.2-μL 6-nm colloidal gold beads (Aurion) was applied to a glow-discharged copper grid coated with 200 mesh holey carbon (Quantifoil R2/1), which was blotted for 2.5 s with a force of 0 before plunge freezing using a Vitrobot Mark IV (Thermo Fisher Scientific) at 4 °C and 100% humidity. The grids were then transferred to a Titan Krios G3i transmission electron microscope (Thermo Fisher Scientific) for either tomography or single particle data collection. The microscope was operated at 300 kV acceleration voltage and equipped with a post-column energy filter (Bio Quantum, Gatan) that was used with a slit width of 20 eV. Both tilt-series (for tomography) and frame-series (for single particle analysis) data were automatically collected using SerialEM-3.8 software package (http://bio3d.colorado.edu/SerialEM/) at a magnification of 53,000×, resulting in a pixel size of 1.68 Å at the specimen level. Images were recorded with a K3 Summit direct electron detector (Gatan) in super-resolution counting mode with a dose rate of 30 e⁻/pixel/s, corresponding to 10.6 e⁻/Å²/s. For tomography, tilt-series were collected in dose-symmetric scheme with a tilt range of −60° to 60° and a step size of 4°. The 0° tilt image was exposed for 2.94 s and fractionated into 90 frames, and the images at other tilt angles were exposed for 0.326 s each and fractionated into 10 frames, which gave a cumulative dose of 135 e⁻/Å² for each tilt series. For single particle analysis, each image was exposed for 5.6 s and fractionated into 28 frames, resulting in a total dose of 60 e⁻/Å². The defocus ranges were approximately −1.5 to −4.0 μm and −1 to −3.5 μm for tilt-series and frame-series datasets, respectively.

### Tomogram reconstruction and subtomogram averaging
Beam-induced motion were corrected with MotionCor2 (ref. 48), and tilt series were aligned with IMOD-4.9 (ref. 49) using the 6-nm colloidal gold as fiducials. The contrast transfer function (CTF) parameters were estimated with emClarity-1.3.0 (ref. 50). Tomograms were reconstructed using IMOD-4.9 (ref. 49) with a binning factor of 4, resulting in a pixel size of 6.72 Å/pixel. SFTSV hexon particles were picked by template matching in emClarity-1.3.0, with the structure of RVFV hexon (EMD-4197) as the template. A total of 27,780 hexon subtomograms were extracted from 118 raw tomograms, which were aligned and classified with 6-fold symmetry applied using emClarity-1.4.3 (ref. 50). A high-quality subset of 15,535 particles was selected after 3D classification, for which the 0° tilt images were extracted from the raw tilt series using emClarity-1.4.3 to reconstruct the final density

map with Relion-4.0 (ref. 51). The resolution was estimated at 10.9 Å by calculating the Fourier shell correlation (FSC) of the two half maps reconstructed by random half sets with a cut-off value of 0.143 (Fig. 1c). The statistics for tomography data collection and image processing are summarized in Supplementary Table 1.

## Single particle image processing and 3D reconstruction

The movie frames were aligned with MotionCor2 (ref. 48) and the CTF parameters were estimated with Gctf-0.50 (ref. 52). A total of 341,154 viral particles were automatically picked from 12,171 micrographs using Gautomatch-0.53 (https://www2.mrc-lmb.cam.ac.uk/download/gautomatch-056/), and extracted from dose-weighted micrographs. Unless otherwise specified, single particle analysis was mainly executed in Relion-3.0 (ref. 53). Several rounds of reference-free 2D classification were performed to clean up the dataset, resulting in a good subset of 118,033 particles. One round of 3D classification by applying I3 symmetry identified one major class with 90,188 good particles, which produced a reconstruction at 6.7 Å resolution after 3D refinement (Supplementary Fig. 1). To overcome the defocus gradient and deformation of the viral particles, block-based reconstruction was performed to further improve the resolution[28]. Three blocks, located at the 5-, 3- and 2-fold axes, were extracted from the final good particle set, producing 5,411,280 sub-particles for each block. An additional round of 3D classification without applying symmetry was performed for each block with local angular search only, resulting in 828,855 good sub-particles for 5-fold block, 2,249,030 sub-particles for 3-fold block, and 1,111,960 sub-particles for 2-fold block. A round of 3D refinement improved the resolution of these blocks to 5.95 Å, 6.19 Å, and 6.17 Å, respectively. For both 3D classification and refinement, particle alignment was focused on a single ASU without applying symmetry. The resulting density maps were then symmetrized to recover the full blocks with 5-, 3- and 2-fold symmetry, respectively. At this stage, CTF refinement was carried out to better estimate the local defocus value for each block. A final round of 3D refinement generated better density maps at resolutions of 4.60 Å, 4.80 Å, and 4.80 Å for the 5-, 3-, and 2-fold blocks, respectively (Supplementary Fig. 1c). For each reconstruction, the resolution was assessed based on the gold-standard criterion of FSC = 0.143, and the local resolution was estimated using ResMap-1.1.4 (ref. 54). In the best resolved regions, the local resolution reached 4.0 Å that allowed the identification of some bulky side chains of amino acid residues (Supplementary Figs. 2 and 3). To further improve the density map for model building, densities of the 12 Gn/Gc heterodimers within an ASU were manually cropped from the locally reconstructed maps of each block and aligned using Chimera-1.11 (ref. 55). As the ectodomain and transmembrane regions display different relative orientations across these quasi-equivalent conformers, the Gn/Gc ectodomain and transmembrane helices were aligned individually and then merged into a composite map using Chimera-1.11. After averaging, the density map was resharpened using the phenix.auto_sharpen program in Phenix-1.20 (ref. 56) with a cut-off resolution of 4.1 Å, which was used for atomic modeling of the Gn/Gc heterodimer. After model refinement, the resolution of this manually averaged map was estimated at 4.3 Å, based on the model v.s. map FSC = 0.5 criterion (Supplementary Fig. 3b).

## Model building, refinement, and visualization

The manually averaged density map for Gn/Gc heterodimer was used for initial model building to resolve the most high-resolution features possible. The X-ray structures of SFTSV Gn (PDB ID: 5Y10) and Gc (PDB ID: 5G47) were first docked into the density map using Chimera-1.11, which were then manually adjusted to improve local fit in Coot-0.8.1 (ref. 57). The C-terminal regions of both proteins were ab-initio built based on structural prediction and map features. The model was refined in real space using Phenix-1.20 (ref. 56) with secondary

structure restraints applied. The stereochemical quality of the atomic model was assessed by MolProbity-4.3.1 (ref. 58), which was used to guide the iterative model adjustment and refinement until convergence. To model the structure of the complete virion, a composite map of the ASU was assembled with the three blocks of locally reconstructed density maps using Chimera-1.11. The model of Gn/Gc heterodimer was symmetrized and then fitted into the ASU map. The model was further refined in real space using Phenix-1.20 following the above-mentioned procedure. The statistics for image processing and model refinement are summarized in Supplementary Table 2. Structural analysis, visualization and figure rendering were performed with Chimera-1.11, ChimeraX-1.5 (ref. 59) or PyMOL-2.0 (https://pymol.org/).

## Conformation clustering of Gn/Gc conformers

The coordinates of the 12 Gn/Gc heterodimers in one ASU were extracted as individual structures and fed into VMD-1.9.3 (ref. 60) software. The structures were aligned with MultiSeq-2.0 (ref. 61) program to calculate $Q_H$ scores that evaluate the homology between each structure pair. A phylogenetic tree was then generated with MultiSeq-2.0 and visualized in FigTree-1.4.4 (http://tree.bio.ed.ac.uk/software/figtree/).

## Pseudovirus packaging assay

The open reading frame of SFTSV M segment, tagged with a C-terminal Flag-peptide (DYKDDDDK), was synthesized by GenScript Biotech (China) and subcloned into pcDNA3.1(+) vector. The N914Q mutant was constructed using the Q5® Site-Directed Mutagenesis Kit (New England Biolab, USA). The WT and mutant pseudoviruses were prepared as previously described[62]. Briefly, HEK293T/17 cells were co-transfected with pLentiCMV-Luc2, pCMV-dR8.2-dvpr and the WT/mutant M expressing plasmids using PEIpro transfection reagent (Polyplus Transfection, France). The cell culture supernatant containing virions was collected at 48 h post transfection, which was clarified by low-speed centrifugation (1000 × $g$, 10 min) and filtered with a 0.45-μm cut-off syringe filter to remove the cell debris. The cleared supernatant was then layered onto a 20% (wt/vol) sucrose cushion and purified by ultracentrifugation (SW41 rotor, 209,900 × $g$, 2 h, 4 °C) using an Optima XPN-100 ultracentrifuge (Beckman Coulter). The resulting pellet was resuspended in sterilized 1×PBS (pH 7.4) and was used for subsequent Western blotting and infection assays. To quantify the viral titer, equal volumes of WT/mutant pseudovirus solutions were used to infect HEK293 cells. At 48 hpi, luciferase activities were determined using the Bright-Lumi™ Firefly Luciferase Assay Kit (Beyotime, China).

## Western blotting

Cells were lysed in ice-cold cell lysis buffer (20 mM Tris pH 7.5, 150 mM NaCl, 1% Triton X-100, and protease inhibitor cocktail), and clarified by centrifugation at 12,000 × $g$ at 4 °C for 5 min. The total protein in the cell extract supernatant was quantified with the Enhanced BCA Protein Assay Kit (Beyotime, China) and resolved by SDS-PAGE. To compare the quantity of WT and mutant pseudovirus particles, the above-mentioned virus samples purified from supernatant were also analyzed. The proteins were transferred onto a nitrocellulose membrane which was blocked with 5% skimmed milk for 1 h at room temperature. The membrane was then probed by appropriate primary antibodies and HRP-conjugated secondary antibodies, and imaged with the chemiluminescence signal produced using the Enhanced ECL Chemiluminescent Substrate Kit (Yeasen Biotechnology, China). For Gn protein detection, a rabbit polyclonal antibody targeting the N-terminal 19 amino acids (anti-HB29 polyclonal antibody, PAB27171, Abnova; Used with a 1:500 dilution) was used as the primary antibody. The Gc protein was detected by a mouse anti-Flag monoclonal antibody (Clone M2, F3165, Sigma-

Aldrich; Used with a dilution of 1:50,000), and β-actin was detected with a mouse monoclonal antibody (66009-1-Ig, Proteintech; Used with a dilution of 1:2,000). Horseradish peroxidase (HRP)-conjugated goat-anti-rabbit IgG (A0208) and goat-anti-mouse IgG (A0216) were purchased from Beyotime Biotech. Inc., and were used with a dilution of 1:2,000. All antibodies were diluted in 1 × Tris-buffered saline (20 mM Tris, pH 7.4, 150 mM NaCl) supplemented with 0.1% Tween-20 and 1% bovine serum albumin (BSA) in all experiments.

## Reporting summary

Further information on research design is available in the Nature Portfolio Reporting Summary linked to this article.

## Data availability

The cryo-EM density maps and associated model coordinates generated in this study have been deposited to the Electron Microscopy Data Bank (EMDB) and Protein Data Bank (PDB) databases, respectively. The accession codes are listed as follows: EMD-35152 (Full virion icosahedral reconstruction), EMD-35176 (2-fold block local reconstruction), EMD-35177 (3-fold block local reconstruction), EMD-35178 (5-fold block local reconstruction), EMD-35183 and 8I4T (composite map of the ASU and the atomic model), EMD-35540 and 8ILQ (locally averaged density of Gn/Gc heterodimer and the atomic model). The subtomogram averaging map of the hexon was deposited as an additional map under the same entry of the ASU map. Source data are provided in this paper.

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

## Acknowledgements

This work was funded by the Science and Technology Program of Guangdong Province (2021B1212030015, Q.L.), the Special Fund for Scientific Innovation Strategy-Construction of High Level Academy of Agriculture Science-Distinguished Scholar (R2020PY-JC001, M.L.), the National Natural Science Foundation of China (31972719, C.L.), Chinese Academy of Medical Sciences Innovation Fund for Medical Sciences (2020-I2M-5-001, N.J.), the Pearl River Talent Plan in Guangdong Province (2019CX01N111, Q.L.), the Scientific and Technological Research Projects of Guangzhou in China (202103000008, Q.L.), and the Medical Innovation Team Project of Jilin University (2022JBGS02, Q.L.).

## Author contributions

S.D., C.L., Q.L., and M.L. conceived the study. S.D., R.P., N.J., J.Q., Q.L., M.L., and C.L. designed the study. S.D., W.X., X.Q., Y.W., and Jiamin W. carried out cell-culture work, virus experiments and prepared and characterized virus samples with supervision from W.L. and Q.L. S.D. and C.L. collected cryo-EM data. S.D., R.P., J.Q., Q.L., and C.L. processed cryo-EM data and made figures. L.L., M.T., G.W., X.S., and Y.M. helped with reagent preparation. Y.G., Jigang W., F.L., M.S., Z.W., H.L., and L.D. helped with virus purification and characterization. S.D., R.P., and C.L. prepared the initial draft and managed the project. S.D., R.P., J.Q., Q.L., and C.L. wrote the manuscript with input from all authors. N.J., Q.L., M.L., and C.L. obtained funding. All authors contributed to discussion and manuscript editing.

## Competing interests

The authors declare no competing interests.
