## [Peer Review File · Nature Communications]

Cryo-EM structure of severe fever with thrombocytopenia syndrome virusREVIEWER COMMENTS

Reviewer #1 (Remarks to the Author):

Du and colleagues report here on the structure of severe fever with thrombocytopenia syndrome virus and organisation of Gn and Gc present on envelop protein. The structure shows that Gc/Gn are presented in heterodimeric form which assemble into pentameric and hexameric peplomer in icosahedral symmetry using SPA. Detailed analysis showed structure of Gn and Gc, basis of Gn/Gc interaction within heterodimer, pentamer and hexamer as well as demonstrated the interaction between pentamer/hexamer. Detailed analysis indicate that there are multiple histidine sites, suggesting to play roles in membrane fusion at acidic environment. Fusion loops are hidden and well protected. Gn/Gc interacts at head and stem domains.. Peplomers are associated via Gc/Gn interaction through N914 on Gc. Comparison with other class II fusion proteins show SFTSV virion structure is different to others. The authors also suggest that some neutralising antibodies targeting at fusion loop site which is hidden at prefusion state. These data are of great interest to the Phlebovirus community, as well as groups focusing on structure-based development of treatment/vaccine options.

Overall, the manuscript is very straightforward, and the structures will certainly prove to be useful for any future vaccine immunogen design work.

However, the following comments should be addressed prior to publication:

1 – The methods section require anyone to be able to reproduce these data and this section needs to be much more detailed. For ex.

- the concentration used in cryoEM/ buffer used, pH? Please provide more details

- Cryo-EM model validation: please provide the Ramachandran Z-scores

2 – Fig. 1 .A showing the tomogram slides, it seems like there is protrusion structure from the virion. Were those particles excluded from analysis?

3. – Fig. 3e. Authors described here five-Beta sheet, what are the distances and amino acids in the region described, would be good to demonstrate.

4. Line 175 Fig. 3i, authors mentioned about F699, in pre vs post fusion. Please also discuss about W652 between pre/post.

5. Line 181, what is the counterpart amino acid(s) of H870 forming interaction as described, please also show the distance.

6. Line 208, authors mentioned about salt bridges and pi-cation interactions – which need to be specified to which amino acid(s) they were referring to.
7. Line 235, Authors suggested that N-linked glycan usually is associated with more than one polysaccharide chain, albeit observed only one in the density map. Is there any evidence demonstrated this claim for SFTSV ? If so, please cite accordingly.
8. Authors used pseudovirus bearing N914Q mutation to demonstrated that this mutation is critical for virion assembly. However, the suggestion should be reworded / reconsidered..5f showed that both virions have similar amounts of proteins. In fig.6g showed that infectivity of the mutant are lower. These two evidence show that the virion assembly is relatively similar, but the virus infectivity is affected by this mutation (either through receptor/ premature inactivated virion). In fact the Gn protein in N914Q mutant is relatively expressed more based on WB data. Also, no internal control for pseudotype virus was included in this experiment, hence it is difficult to quantify the viral output generated from pseudotype and compare between wt and mutant. Therefore, care should be taken on this point.
9. Line 295, fig. S7. Authors described about F699 and depicted in fig.S7. , is it in -pre or post fusion conformation?
10. Please also show local resolution on Fig. 1 (Gn/Gc map), is that the map from hexamer or pentamer or combined? Please describe.

Reviewer #2 (Remarks to the Author):

Du et al. present a nice story of a segmented, negative-strand RNA virus, the severe fever with thrombocytopenia syndrome virus (SFTSV) and its 2 membrane proteins, glycoproteins Gn/Gc. The authors solved the cryo-EM structure of highly dynamic SFTSV by using a block-based reconstruction method and identified that Gn/Gc exists as a heterodimer. After that, EM densities of the Gn/Gc heterodimer, which is the basic structural unit, were manually averaged to facilitate atomic modeling for the virion. Then with an emphasis on understanding how the fusion loops of Gc are shielded and maintained in the metastable prefusion conformation, the authors described the structures, organizations, and interactions of Gn and Gc in the context of heterodimer, pentameric and hexameric peplomers that the heterodimers form, as well as in between the peplomers. The authors proposed a model of SFTSV hexameric peplomers conformational changes that could mediate membrane fusion.

This manuscript reports the highest resolution structure of SFTSV to date that suggests mechanism of the virus' host cell membrane fusion and infection, while there are also some issues within the body text and data interpretation. I would recommend its acceptance for publication if concerns listed below could be addressed:

One issue is that the authors should clarify clearly which results are / are not from this study and better arrange the Results and Discussion throughout the manuscript. For example, the Results part titled "Conformational changes of Gc for membrane fusion" are comparisons between structures in this study

and literature and speculation based on structural analysis. The authors may change the title or move this part to the Discussion. In Fig 3j, the authors should label the published post-fusion conformation with its PDB ID.

Minor issues:

1.Line 97, the authors may explain why were the hexagonal, not the pentagonal particles extracted to improve the resolution

2.Line 103, "Supplementary Fig. 1" should be changed to "Supplementary Fig. 1c"

3.Line 106, "Fig. 1c and d" should be changed to "Supplementary Fig. 1c and d"

4.Line 112, in figure 2 there is no information about the triangulation number information, the author may add it

5.Line 118, the "Gn head domain" may be changed to "Gn head" since the authors already used the same word "domain" to describe the individual 3 domains of the head

6.Line 123, "of Gc" should be added after "(TM) domain"

7.Line 133, "Fig. 2b and c" should be changed to "Fig. 3b and c"

8.Line 144, the authors should explain why the structure is a compressed conformation to whom doesn't know this field well and may add references 22 and 30 here

9.Line 148, "Supplementary Fig. 4" should be changed to "Supplementary Fig. 4e and g"

10.In figure 3f, Gn residues may be labeled to help the reader find where this part is in the whole Gn structure

11.Line 159 and in figure 3, the long membrane proximal linkers of Gn was not indicated, only the one for Gc was

12.Line 171, please add reference after "...alphavirus E proteins"

13.Line 173, add "3e" to "Fig. 3h"

14.Line 193, "Fig. 4" should be "Fig. 4a and h"

15.Line 194, the authors should specify that which type of hexon is being analyzed here

16.Line 200, the authors should label the domain II of the adjacent Gc subunit in Fig. 4e

17.Line 206 and fig. 4g, the authors may not label the residues that are not negatively charged and aromatic

18.Line 227, "Fig. 5" should be changed to "Fig. 5a"

19.Line 274, "prior to" should be changed to "prone to"

20.Line 288 and fig. 7a, the cap helix in Gn domain C that's targeted by Mab4-5 was not label in the figure

21. Line 293, the authors should explain the docking method
22. Line 315, reference for the Reed-Muench method should be included
23. Line 327, this title should be changed to include the cryo-ET part
24. Line 361, reference 42 is the same with the reference 38
25. Line 539, reference 35 cannot be found by searching "Science 360, 48-+, doi:ARTN eao728310.1126/science.aao7283 (2018)"
26. Fig. 1b, the authors should explain that the left and right half are the same
27. Line 616, Fig. 2a, "by radius" should be changed to "by the radial distance from the virion center"
28. Fig. 3c, the 2 dashed line boxes should be mentioned for the fig. 3e
29. Fig. 5a, the 2 dashed line boxes should be mentioned for the fig. b and c; the authors should also indicate the boxes in fig. a for the fig. d and e
30. Supplementary Fig. 1, the authors should label more clearly which one is 5-fold, 2-fold, and 3-fold block respectively for the 3 parts in fig. d
31. Line 715, add "on" after H407
32. Line 745, it should be specified that the ASU has the same number code with the one in fig. 2d
33. Line 768, percentage of the disallowed is too high, the authors should try to further optimize the structure

Reviewer #3 (Remarks to the Author):

The manuscript by Du et al describes the 4.3 Å resolution cryo-EM structure of the virion of the "severe fever with thrombocytopenia virus" (SFTSV), a member of the Phenuiviridae family in the order Bunyavirales. This family contains a number of human pathogenic arboviruses, some transmitted by ticks, such as SFSTV, others transmitted by insects, like the Rift Valley fever virus (mosquito borne) and Toscana virus (transmitted by sandflies). Global warming is making such arthropod-borne viruses cause outbreaks in more and more extended regions of the planet, and are becoming a serious concern for public health. The manuscript describes the detailed interactions of the two envelope glycoproteins, Gn and Gc, which form a heterodimer that makes additional quaternary interactions to constitute the virus particle, forming of pentameric and 110 hexameric "peplomers" that completely enclosed the viral membrane to make the virion. They show that, in addition to the external hydrophobic shell, a C-terminal alpha helix of Gn makes an internal scaffold underneath the viral membrane. This helix is well ordered under most of the hexameric peplomers, but not under the pentameric ones. The paper also describes in detail the various contacts between the 720 Gn/Gc heterodimers, which are the building blocks of the peplomers, as well as inter-peplomer interactions to cement the viral particle.

The structure reported will serve as solid basis to understand the epitopes of neutralizing antibodies in the context of the infectious particles, showing that some are buried and require “particle breathing for exposure”, so this structure is an important landmark in identifying ways to combat pathogens of the whole Phenuivirus family. Technically, the structures are of high quality and the resolution, although limited to 4.3Å, provides essential elements to understand the relevant interactions making up the particle.

I only have one issue, and it regards the domain labels. What the authors call “stem”, is actually the third Ig-like domain present in Gn and in alphavirus Gc (which they display in Fig. 6f and 6g). Using the term “stem” is unfortunate, because for class II viruses this term is used to the flexible segments of Gn and Gc (Or of E2 and E1 for alphaviruses) connecting to the TM segment. They are, therefore, downstream the domain called “stem”. Moreover, this domain corresponds to domain “C”, as accepted for the accompanying protein of class II viruses. In addition, what the authors called “domain B” of Gn appears to correspond to the “beta-ribbon” originally identified in alphaviruses, and which organizes the three domains (A, B and C) of the accompanying protein, Gn or E2, depending on the virus, even though in SFTSV the beta-ribbon is more elaborated, with additional secondary structure elements. What the authors term “Domain C” of SFTSV Gn indeed corresponds to domain B in the other class II proteins, and I therefore strongly suggest that the authors stick to the accepted nomenclature, renaming their domain B into something like beta connector or similar, and renaming the stem domain as domain C.

In relation to the above point, in alphaviruses and in hantaviruses, domain C (the domain called here “stem”), is what cements each spike, being responsible for the most of the intra-spike interactions, whereas the membrane fusion protein (Gc in bunyaviruses and E1 in alphaviruses) is responsible for the lateral, spike-spike interactions. In SFTSV this does not seem to be the case, and it is important that the authors better discuss it, as it is not clear in the text. Also, it is not clear what the authors call a spike, is it synonymous of “peplomer”? Maybe they should stick to one single term, to avoid confusion.

REVIEWER COMMENTS

Reviewer #1 (Remarks to the Author):

Du and colleagues report here on the structure of severe fever with thrombocytopenia syndrome virus and organisation of Gn and Gc present on envelop protein. The structure shows that Gc/Gn are presented in heterodimeric form which assemble into pentameric and hexameric peplomer in icosahedral symmetry using SPA. Detailed analysis showed structure of Gn and Gc, basis of Gn/Gc interaction within heterodimer, pentamer and hexamer as well as demonstrated the interaction between pentamer/hexamer. Detailed analysis indicate that there are multiple histidine sites, suggesting to play roles in membrane fusion at acidic environment. Fusion loops are hidden and well protected. Gn/Gc interacts at head and stem domains. Peplomers are associated via Gc/Gn interaction through N914 on Gc. Comparison with other class II fusion proteins show SFTSV virion structure is different to others. The authors also suggest that some neutralising antibodies targeting at fusion loop site which is hidden at prefusion state. These data are of great interest to the Phlebovirus community, as well as groups focusing on structure-based development of treatment/vaccine options. Overall, the manuscript is very straightforward, and the structures will certainly prove to be useful for any future vaccine immunogen design work. However, the following comments should be addressed prior to publication:

Response: Thank you for the positive comment and the constructive suggestions for improving the manuscript! We have revised our manuscript according to all the issues listed below.

1 – The methods section require anyone to be able to reproduce these data and this section needs to be much more detailed. For ex.

- the concentration used in cryoEM/ buffer used, pH? Please provide more details

- Cryo-EM model validation: please provide the Ramachandran Z-scores

Response: We have added more method details as suggested. The Ramachandran Z-scores have been added to Supplementary Table 2.

2 – Fig. 1 .A showing the tomogram slides, it seems like there is protrusion structure from the virion. Were those particles excluded from analysis?

Response: We believe the protrusion structures are actually the penton and hexon peplomers in the side views, which was therefore also described as spikes (unified as peplomers in the revised version as suggested by reviewer #3). In some regions, probably because of virion deformation, these protrusion features may look a bit different. For subtomogram averaging, we used the published structure of RVFV hexon as a reference to pick particles by template-matching. Therefore, it is foreseeable that the distorted or low contrast particles would be sorted out in subsequent classification steps.

3. – Fig. 3e. Authors described here five-Beta sheet, what are the distances and amino acids in the region described, would be good to demonstrate.

Response: As suggested, we have added the description about contacting residues in this region. Specifically, the b strand of Gc (residues 646-648) pairs with the edge strand of Gn β -connector domain (residues 245-247), so that connecting the c-d-b strands of Gc and two strands of Gn to create a 5-strand β -sheet. The interacting residues and the hydrogen bonds are shown in Fig. 3e, which are within 3.5 Å distance, the typical cut-off value for hydrogen bond interactions. The interaction is mainly mediated by the main chains, and also involves the side chain of R648.

4. Line 175 Fig. 3i, authors mentioned about F699, in pre vs post fusion. Please also discuss about W652 between pre/post.

Response: As suggested, we have added the description about the conformational change of W652 before and after membrane fusion. Residue W652 slightly moves downwards, together with the entire b-c loop, in the post-fusion conformation. This is possibly a result of concerted conformational change to create space for accommodating the erected c-d loop.

5. Line 181, what is the counterpart amino acid(s) of H870 forming interaction as described, please also show the

distance.

Response: The main chain of H870 possibly interacts with K409, and its side chain potentially forms two hydrogen bonds with S372 and T374. Because the resolution of our structure is somehow limited, we do not want to over-claim the details of atomic interactions. We were trying to be conservative in our language for these descriptions. Also, because of this uncertainty, we took the residues within 5 Å distance into consideration to analyze the potential interactions.

6. Line 208, authors mentioned about salt bridges and pi-cation interactions – which need to be specified to which amino acid(s) they were referring to.

Response: We have specified the interacting residues in the revised manuscript. Gn residues E201 and E193 of Gn may potentially form two salt bridges with R895 and K785 in Gc, respectively, and residues E189, Q199 and S200 of Gn likely interact with S740, C586 and R895 of Gc via hydrogen bonds (Fig. 4g). The side chain of F197 is actually not involved in the interaction. We have removed it from the figure and text. Again, we were trying to be conservative in our language to avoid over-interpretation.

7. Line 235, Authors suggested that N-linked glycan usually is associated with more than one polysaccharide chain, albeit observed only one in the density map. Is there any evidence demonstrated this claim for SFTSV? If so, please cite accordingly.

Response: Sorry for the ambiguity on this point! To our knowledge, there is no direct evidence demonstrating the glycosylation on SFTSV. However, our structure actually provides some clues for that. In our structure, we only modeled one sugar moiety on N914, but we did observe some additional density that may represent a polysaccharide chain. Indeed, the density is not strong enough to be modeled with confidence, possibly because of the flexible conformation of glycans. We have updated the figure (Fig. 3d) and descriptions accordingly. The point is that a polysaccharide chain may contribute more interactions with adjacent subunits than those shown in the model, thus possibly playing a more important role for viral assembly. This is supported by the mutagenesis study showing the severely impaired viral assembly in the absence of this glycan modification.

8. Authors used pseudovirus bearing N914Q mutation to demonstrated that this mutation is critical for virion assembly. However, the suggestion should be reworded / reconsidered..5f showed that both virions have similar amounts of proteins. In fig.6g showed that infectivity of the mutant are lower. These two evidence show that the virion assembly is relatively similar, but the virus infectivity is affected by this mutation (either through receptor/ premature inactivated virion). In fact the Gn protein in N914Q mutant is relatively expressed more based on WB data. Also, no internal control for pseudotype virus was included in this experiment, hence it is difficult to quantify the viral output generated from pseudotype and compare between wt and mutant. Therefore, care should be taken on this point.

Response: Thanks for the comment! First, we wish to clarify that the Western blot was showing the protein expression level in cells, to demonstrate the mutation did not affect the protein production. On that condition, we observed the viral titer of mutant became much lower compared to the WT, suggesting the efficiency of virion assembly was affected by this mutation. However, we agree that this could not rule out the possibility of impairing viral receptor binding or other steps of infection. It is not sufficient to demonstrate the direct contribution of the glycans to the assembly of viral particles. To further investigate this question, we also quantified the Gn/Gc proteins in purified pseudotyped virions. Both the WT and mutant pseudoviruses were packaged in 293T cells and the supernatant was collected and subjected to ultracentrifugation to pellet the viral particles, following the same procedure for purifying virions for cryo-EM studies. Even though the mutant protein is expressed at a similar level to the WT in cells, the quantity of Gn/Gc in mutant virions is significantly lower than the WT, suggesting fewer mutant virions being produced. Consistent with this evidence, the mutant virions displayed a much-reduced infectivity than the WT virions. Together, these observations demonstrate the N914Q substitution remarkably reduced the efficiency of virion assembly, suggesting an important role of the glycans for stabilizing the structure of viral particles.

9. Line 295, fig. S7. Authors described about F699 and depicted in fig.S7. , is it in -pre or post fusion conformation?

Response: It is in the prefusion conformation. The structure of MAb4-5 in complex with Gn head was superimposed

with the Gn/Gc heterodimer resolved in this study by aligning the Gn head, so that the binding mode of MAb4-5 in the context of virion could be modeled. To avoid confusion, we have modified the text to explicitly describe the “Gn/Gc heterodimer in the prefusion conformation” (lines 312-316).

10. Please also show local resolution on Fig. 1 (Gn/Gc map), is that the map from hexamer or pentamer or combined? Please describe.

Response: The Gn/Gc map was generated by averaging the 12 heterodimers within an ASU together. Because the TM region shows variable conformation, the ectodomain and the TM regions were individually aligned and averaged, which were then combined to produce a composite map. We have added the details to the Methods section. And the local resolution map was added and reorganized as Supplementary Fig. 2.

Reviewer #2 (Remarks to the Author):

Du et al. present a nice story of a segmented, negative-strand RNA virus, the severe fever with thrombocytopenia syndrome virus (SFTSV) and its 2 membrane proteins, glycoproteins Gn/Gc. The authors solved the cryo-EM structure of highly dynamic SFTSV by using a block-based reconstruction method and identified that Gn/Gc exists as a heterodimer. After that, EM densities of the Gn/Gc heterodimer, which is the basic structural unit, were manually averaged to facilitate atomic modeling for the virion. Then with an emphasis on understanding how the fusion loops of Gc are shielded and maintained in the metastable prefusion conformation, the authors described the structures, organizations, and interactions of Gn and Gc in the context of heterodimer, pentameric and hexameric peplomers that the heterodimers form, as well as in between the peplomers. The authors proposed a model of SFTSV hexameric peplomers conformational changes that could mediate membrane fusion.

This manuscript reports the highest resolution structure of SFTSV to date that suggests mechanism of the virus' host cell membrane fusion and infection, while there are also some issues within the body text and data interpretation. I would recommend its acceptance for publication if concerns listed below could be addressed:

Response: Thanks for the positive comments and suggestions! We have intensively revised the manuscript to address all the issues listed below.

One issue is that the authors should clarify clearly which results are / are not from this study and better arrange the Results and Discussion throughout the manuscript. For example, the Results part titled “Conformational changes of Gc for membrane fusion” are comparisons between structures in this study and literature and speculation based on structural analysis. The authors may change the title or move this part to the Discussion. In Fig 3j, the authors should label the published post-fusion conformation with its PDB ID.

Response: Thanks for the suggestion! We have thoroughly reorganized the text accordingly. In the section describing Gn/Gc heterodimer structure, we kept the structural comparison of pre- and post-fusion conformations, by solely describing the structural differences. The description about the model of membrane fusion has been moved to the Discussion section. In Fig. 3j, the PDB ID for the postfusion structure has been added to the figure.

Minor issues:

1.Line 97, the authors may explain why were the hexagonal, not the pentagonal particles extracted to improve the resolution

Response: This is because there are more hexagonal particles than pentagonal ones in each virion, so more likely to achieve higher resolution. It was a pilot analysis though to test the attainable resolution using the hexagonal particles, and to produce an initial model for further analysis. Since we obtained about 1 nm resolution for hexagonal particles, we did not further process the pentagonal particles which were foreseeably unable to produce a better reconstruction. Therefore, we turned to single particle analysis approach to pursue a near-atomic resolution reconstruction. We have modified the text accordingly (lines 95-98).

2.Line 103, “Supplementary Fig. 1” should be changed to “Supplementary Fig. 1c”

Response: Corrected.

3.Line 106, “Fig. 1c and d” should be changed to “Supplementary Fig. 1c and d”

Response: Corrected.

4.Line 112, in figure 2 there is no information about the triangulation number information, the author may add it

Response: As suggested, the triangulation information has been added to Fig. 2a (H=2, K=2).

5.Line 118, the “Gn head domain” may be changed to “Gn head” since the authors already used the same word “domain” to describe the individual 3 domains of the head

Response: Corrected.

6.Line 123, “of Gc” should be added after “(TM) domain”

Response: The intramembrane (IM) helix is actually in the C-terminus of Gn, downstream of the TM helix.

7.Line 133, “Fig. 2b and c” should be changed to “Fig. 3b and c”

Response: Corrected.

8.Line 144, the authors should explain why the structure is a compressed conformation to whom doesn't know this field well and may add references 22 and 30 here

Response: Thanks for the suggestion. We have added a few sentences describing this point. And references 22 and 30 were cited accordingly (lines 143-146).

9.Line 148, “Supplementary Fig. 4” should be changed to “Supplementary Fig. 4e and g”

Response: Corrected.

10.In figure 3f, Gn residues may be labeled to help the reader find where this part is in the whole Gn structure

Response: As suggested, we have added labels for the key interacting residues of Gn in the figure.

11.Line 159 and in figure 3, the long membrane proximal linkers of Gn was not indicated, only the one for Gc was

Response: We have added two additional panels in Supplementary Fig. 4b and 5b to show the overall structure of Gn and Gc, respectively. The membrane-proximal linkers are highlighted in the figures.

12.Line 171, please add reference after “...alphavirus E proteins”

Response: Corrected.

13.Line 173, add “3e” to “Fig. 3h”

Response: Corrected.

14.Line 193, “Fig. 4” should be “Fig. 4a and h”

Response: Corrected.

15.Line 194, the authors should specify that which type of hexon is being analyzed here

Response: Basically, the structure of the three types of hexons are very similar, with only some local variations. The descriptions about the assembly of hexons can actually be applied to all three types. For clarity, we have added “with hexon C as an example”.

16.Line 200, the authors should label the domain II of the adjacent Gc subunit in Fig. 4e

Response: Corrected.

17.Line 206 and fig. 4g, the authors may not label the residues that are not negatively charged and aromatic

Response: As suggested here (as well as reviewer #1), we have added the potential contacting residues to show hydrogen bonds and salt bridges. The text has also been updated accordingly.

18.Line 227, “Fig. 5” should be changed to “Fig. 5a”

Response: Corrected.

19.Line 274, “prior to” should be changed to “prone to”

Response: Corrected.

20.Line 288 and fig. 7a, the cap helix in Gn domain C that’s targeted by Mab4-5 was not label in the figure

Response: Corrected.

21.Line 293, the authors should explain the docking method

Response: We have added the description about docking. The structure of MA4-5 in complex with Gn head was superimposed with the Gn/Gc heterodimer by aligning the Gn head.

22.Line 315, reference for the Reed-Muench method should be included

Response: Corrected.

23.Line 327, this title should be changed to include the cryo-ET part

Response: Corrected.

24.Line 361, reference 42 is the same with the reference 38

Response: Corrected.

25.Line 539, reference 35 cannot be found by searching “Science 360, 48+, doi:ARTN eao728310.1126/science.aao7283 (2018)”

Response: Corrected.

26.Fig. 1b, the authors should explain that the left and right half are the same

Response: We have modified the legend accordingly.

27.Line 616, Fig. 2a, “by radius” should be changed to “by the radial distance from the virion center”

Response: Corrected.

28.Fig. 3c, the 2 dashed line boxes should be mentioned for the fig. 3e

Response: We have modified the legend accordingly.

29.Fig. 5a, the 2 dashed line boxes should be mentioned for the fig. b and c; the authors should also indicate the boxes in fig. a for the fig. d and e

Response: We have modified the figure and legend accordingly.

30.Supplementary Fig. 1, the authors should label more clearly which one is 5-fold, 2-fold, and 3-fold block respectively for the 3 parts in fig. d

Response: The label has been updated and these panels have been reorganized as Supplementary Fig. 2.

31.Line 715, add “on” after H407

Response: Corrected.

32.Line 745, it should be specified that the ASU has the same number code with the one in fig. 2d.

Response: Corrected.

33.Line 768, percentage of the disallowed is too high, the authors should try to further optimize the structure

Response: We have further refined the atomic model. The new version has 0.2% outliers (2 residues).

Reviewer #3 (Remarks to the Author):

The manuscript by Du et al describes the 4.3 Å resolution cryo-EM structure of the virion of the “severe fever with thrombocytopenia virus” (SFTSV), a member of the Phenuiviridae family in the order Bunyavirales. This family contains a number of human pathogenic arboviruses, some transmitted by ticks, such as SFSTV, others transmitted by insects, like the Rift Valley fever virus (mosquito borne) and Toscana virus (transmitted by sandflies). Global warming is making such arthropod-borne viruses cause outbreaks in more and more extended regions of the planet, and are becoming a serious concern for public health. The manuscript describes the detailed interactions of the two envelope glycoproteins, Gn and Gc, which form a heterodimer that makes additional quaternary interactions to constitute the virus particle, forming of pentameric and 110 hexameric “peplomers” that completely enclosed the viral membrane to make the virion. They show that, in addition to the external hydrophobic shell, a C-terminal alpha helix of Gn makes an internal scaffold underneath the viral membrane. This helix is well ordered under most of the hexameric peplomers, but not under the pentameric ones. The paper also describes in detail the various contacts between the 720 Gn/Gc heterodimers, which are the building blocks of the peplomers, as well as inter-peplomer interactions to cement the viral particle.

The structure reported will serve as solid basis to understand the epitopes of neutralizing antibodies in the context of the infectious particles, showing that some are buried and require “particle breathing for exposure”, so this structure is an important landmark in identifying ways to combat pathogens of the whole *Phenuivirus* family. Technically, the structures are of high quality and the resolution, although limited to 4.3Å, provides essential elements to understand the relevant interactions making up the particle.

Response: Thank you for acknowledging our work!

1. I only have one issue, and it regards the domain labels. What the authors call “stem”, is actually the third Ig-like domain present in Gn and in alphavirus Gc (which they display in Fig. 6f and 6g). Using the term “stem” is unfortunate, because for class II viruses this term is used to the flexible segments of Gn and Gc (Or of E2 and E1 for alphaviruses) connecting to the TM segment. They are, therefore, downstream the domain called “stem”. Moreover, this domain corresponds to domain “C”, as accepted for the accompanying protein of class II viruses. In addition, what the authors called “domain B” of Gn appears to correspond to the “beta-ribbon” originally identified in alphaviruses, and which organizes the three domains (A, B and C) of the accompanying protein, Gn or E2, depending on the virus, even though in SFTSV the beta-ribbon is more elaborated, with additional secondary structure elements. What the authors term “Domain C” of SFTSV Gn indeed corresponds to domain B in the other class II proteins, and I therefore strongly suggest that the authors stick to the accepted nomenclature, renaming their domain B into something like beta connector or similar, and renaming the stem domain as domain C.

Response: Thanks for the suggestion. We have renamed the Gn domain B as β -connector domain, and renamed domain C as domain B, and renamed stem as domain C. The capping element of Hantavirus was renamed as cap-loop. The β -ribbon connector of Hantavirus Gn is more integrated into the large domain B. For clarity, we did not further divide it out, but leave it as a part of domain B, to keep the 3-domain architecture of Gn head (Fig. 6). We also added some discussions about the comparison of these structures (lines 255-261).

2. In relation to the above point, in alphaviruses and in hantaviruses, domain C (the domain called here “stem”), is what cements each spike, being responsible for the most of the intra-spike interactions, whereas the membrane fusion protein (Gc in bunyaviruses and E1 in alphaviruses) is responsible for the lateral, spike-spike interactions. In SFTSV this does not seem to be the case, and it is important that the authors better discuss it, as it is not clear in the text. Also, it is not clear what the authors call a spike, is it synonymous of “peplomer”? Maybe they should stick to

one single term, to avoid confusion.

Response: As suggested, we have added some discussions about the different intra- and inter-peplomer interactions of these viruses (lines 262-274). Yes, spike is synonymous of peplomer, as the peplomers usually create spiky features on virion surface. For alphaviruses and hantaviruses, people have been using “spikes” to describe the high-order assembly of glycoproteins. For phenuiviruses, it is more common to use peplomers or capsomers. However, we do not want to confuse readers about “capsomers” that make up the nucleocapsid of many DNA viruses with the “peplomers” that describe the glycoprotein spikes on the viral envelope, so we don’t use capsomers in the text. As suggested, we have unified the description about phenuivirus glycoprotein spikes as “peplomers” throughout the text.

REVIEWERS' COMMENTS

Reviewer #1 (Remarks to the Author):

All the questions are fulfilled and addressed and no further questions/suggestions from me.

Reviewer #2 (Remarks to the Author):

In this version of manuscript, the authors have substantially revised the manuscript, including modifying/ reorganizing the text and figures, and correcting all the figures' references. Results from this study are well indicated in figures. Gn/Gc heterodimer structure model is refined.

Overall, this work is of significance to the bunyavirus field and a broader virology community in terms of mechanisms of viral assembly and host infection. Particularly, the authors revealed a novel Gn/Gc pre-fusion conformation that shields the fusion loop on Gc by cryo-EM single particle analysis. I would like to recommend it to be accepted for publication.

Please address the following comment before publication:

1. Caption for fig 3g is missing, please also try to indicate the long linker loop in fig 3g

Reviewer #3 (Remarks to the Author):

The authors have done a good job incorporating my suggestions as well as those from the other reviewers. The manuscript is clear, and the work presented is a real tour de force. These nanomachines being so flexible and difficult to capture at high resolution.

The only part that is still not clear to me is the description of the inter-peplomer contacts, perhaps because Figure 5a is still difficult to read. If understand correctly, the contacts in the TM regions involve three rings (or three peplomers), but at the level of the Gn/Gc ectodomains (or "the trunk") its binary, involving two peplomers facing each other in each case? And in this case, two Gn/Gc protomers of one peplomer interact with two protomer in front, making type1 and type-2 contacts (as defined in Figure 5)?

It would be useful to have a Figure with only the domains Gn(C) and Gc(I-III) of the protomers in adjacent peplomers were displayed.

I have a comment on the discussion, the sentence saying (line 312) “indicating the binding of MAb4-5 to SFTSV virions mainly occurs in the endosome”. But if binding occurs mainly in the endosome, what brings enough antibody into the endosome together with the virus particle to neutralize it? This needs some explanation.

As the authors are non-native English speakers, there are still some issues with the writing. I list a few instances:

Line 118: “In both pentons and hexons, the Gn head clusters together”, should be the “the Gn heads cluster together”

Line 136: “...in hantavirus is a β -strand hairpin (named the cap loop) in which connecting loop caps fusion loops of Gc in a similar topology”. This sentence is confusing. I think it is meant: “...in hantaviruses, a long beta-hairpin in domain A projects a loop (termed “capping loop”) which caps the Gc fusion loop”

Line 176: “the positive charge will force histidines to expel other basic residues’, repel, rather than “expel”.

Line 183 “...Gn/Gc heterodimers, as basic structural unit, intercalate to each other” Intercalate with each other. But I would just say that they make lateral interactions to make rings of 5 or 6.

In supplementary Figure 7C, is hexon C represented at the same scale than E and P?

Line 214: “Structure based phylogentic analysis”. The term “phylogenetic” does not apply here, since it is not comparing Hn/Gc heterodimers from different organisms, but rather different conformations observed for the same molecule. Although the representation is the analysis is the same, the result is a classification of the conformations into groups, but not a phylogenetic tree.

Line 309: ‘implying the “breath”of virions that transiently exposes the epitope’ Implying breathing of the virion.

The authors want to thank all the reviewers for their time and efforts to review this work and for their constructive comments and suggestions to improve the manuscript!

Reviewer #1 (Remarks to the Author):

All the questions are fulfilled and addressed and no further questions/suggestions from me.

Reviewer #2 (Remarks to the Author):

In this version of manuscript, the authors have substantially revised the manuscript, including modifying/ reorganizing the text and figures, and correcting all the figures' references. Results from this study are well indicated in figures. Gn/Gc heterodimer structure model is refined.

Overall, this work is of significance to the bunyavirus field and a broader virology community in terms of mechanisms of viral assembly and host infection. Particularly, the authors revealed a novel Gn/Gc pre-fusion conformation that shields the fusion loop on Gc by cryo-EM single particle analysis. I would like to recommend it to be accepted for publication.

Please address the following comment before publication:

1. Caption for fig 3g is missing, please also try to indicate the long linker loop in fig 3g

Response: Thanks for the comment! We have added the caption accordingly. The long linker loop is labeled with "Gc-linker" in the figure.

Reviewer #3 (Remarks to the Author):

The authors have done a good job incorporating my suggestions as well as those from the other reviewers. The manuscript is clear, and the work presented is a real tour de force. These nanomachines being so flexible and difficult to capture at high resolution.

The only part that is still not clear to me is the description of the inter-peplomer contacts, perhaps because Figure 5a is still difficult to read. If understand correctly, the contacts in the TM regions involve three rings (or three peplomers), but at the level of the Gn/Gc ectodomains (or "the trunk") its binary, involving two peplomers facing each other in each case? And in this case, two Gn/Gc protomers of one peplomer interact with two protomer in front, making type1 and type-2 contacts (as defined in Figure 5)?

It would be useful to have a Figure with only the domains Gn(C) and Gc(I-III) of the protomers in adjacent peplomers were displayed.

Response: Sorry for the confusion on this! Actually, both the TM region and ectodomains are involved in interactions between 3 adjacent peplomers to make type 1, type 3 and type 4 interactions. The type 1 interaction is defined at the interface between Gn domain C of one peplomer and the Gc domain III of the neighboring peplomer. Three sets of type 1 interaction create a closed network to lock three adjacent peplomers. For type 2 interaction, it only involves two peplomers. To facilitate reading, we have prepared an additional supplementary figure (Supplementary Fig. 8) to reveal the contexts of each type of interactions as suggested.

I have a comment on the discussion, the sentence saying (line 312) "indicating the binding of MAb4-5 to SFTSV virions mainly occurs in the endosome". But if binding occurs mainly in the endosome, what brings enough antibody into the endosome together with the virus particle to neutralize it? This needs some explanation.

Response: Thanks for the comment! This is a similar scenario to many fusion loop targeting

antibodies of flaviviruses, which may either bind virions during breathing or be internalized into the endosome together with viral particles to interfere with the membrane fusion process. It's conceivable that this type of antibodies may need to reach higher concentrations in the body fluid than other antibodies targeting other fully exposed epitopes, so that to ensure sufficient antibodies being internalized for viral neutralization in the endosome. Thus, these antibodies commonly display high IC50 values and variable neutralizing efficiencies against different viruses which might be associated with different virion stability and thus different conformational heterogeneity among virion populations. For example, the antibody 2A10G6 has been shown capable of neutralizing different types of dengue viruses, YFV, JEV, WNV and Zika virus, with different inhibition efficiencies, by intercepting the later steps of virus entry after receptor binding (Deng et al., 2011, ref. 42; Dai et al., 2016, ref. 40), which suggests the antibody is internalized together with viral particles into the endosome to inhibit viral-host membrane fusion. However, it's not yet known how cells regulate antibody uptake to ensure viral neutralization in the endosome. We have added this point to the discussion.

As the authors are non-native English speakers, there are still some issues with the writing. I list a few instances:

Line 118: "In both pentons and hexons, the Gn head clusters together", should be the "the Gn heads cluster together"

Response: Corrected.

Line 136: "...in hantavirus is a β -strand hairpin (named the cap loop) in which connecting loop caps fusion loops of Gc in a similar topology". This sentence is confusing. I think it is meant: "...in hantaviruses, a long beta-hairpin in domain A projects a loop (termed "capping loop") which caps the Gc fusion loop"

Response: We have rephrased the sentence accordingly.

Line 176: "the positive charge will force histidines to expel other basic residues", repel, rather than "expel".

Response: Corrected.

Line 183 "...Gn/Gc heterodimers, as basic structural unit, intercalate to each other" Intercalate with each other. But I would just say that they make lateral interactions to make rings of 5 or 6.

Response: We have rephrased the sentence accordingly.

In supplementary Figure 7C, is hexon C represented at the same scale than E and P?

Response: Thanks for pointing this out! In the original figure, the hexon C was misrepresented with a smaller scale. We have updated the figure accordingly.

Line 214: "Structure based phylogenetic analysis". The term "phylogenetic" does not apply here, since it is not comparing Hn/Gc heterodimers from different organisms, but rather different conformations observed for the same molecule. Although the representation is the analysis is the same, the result is a classification of the conformations into groups, but not a phylogenetic tree.

Response: Thanks for the suggestion! We have replaced it with "Conformation clustering of

Gn/Gc conformers”.

Line 309: ‘implying the “breath”of virions that transiently exposes the epitope’ Implying breathing of the virion.

Response: Corrected.